# Weak-for-Strong:
# Training Weak Meta-Agent to Harness Strong Executors

**Fan Nie**
Stanford University, USA
niefan@stanford.edu

**Lan Feng**
EPFL, Switzerland
lan.feng@epfl.ch

**Haotian Ye**
Stanford University, USA
haotianye@stanford.edu

**Weixin Liang**
Stanford University, USA
wxliang@stanford.edu

**Pan Lu**
Stanford University, USA
panlu@stanford.edu

**Huaxiu Yao**
UNC-Chapel Hill, USA
huaxiu@cs.unc.edu

**Alexandre Alahi**
EPFL, Switzerland
alexandre.alahi@epfl.ch

**James Zou**
Stanford University, USA
jamesz@stanford.edu

## Abstract

Efficiently leveraging of the capabilities of contemporary large language models (LLMs) is increasingly challenging, particularly when direct fine-tuning is expensive and often impractical. Existing training-free methods, including manually or automated designed workflows, typically demand substantial human effort or yield suboptimal results. This paper proposes Weak-for-Strong Harnessing (W4S), a novel framework that customizes smaller, cost-efficient language models to design and optimize workflows for harnessing stronger models. W4S formulates workflow design as a multi-turn markov decision process and introduces reinforcement learning for agentic workflow optimization (RLAO) to train a weak meta-agent. Through iterative interaction with the environment, the meta-agent learns to design increasingly effective workflows without manual intervention. Empirical results demonstrate the superiority of W4S that our 7B meta-agent, trained with just one GPU hour, outperforms the strongest baseline by 2.9% ∼ 24.6% across eleven benchmarks, successfully elevating the performance of state-of-the-art models such as GPT-3.5-Turbo and GPT-4o. Notably, W4S exhibits strong generalization capabilities across both seen and unseen tasks, offering an efficient, high-performing alternative to directly fine-tuning strong models. Code is available here.

## 1 Introduction

Despite the rapid advancement of large language models (LLMs) such as GPT-4o (OpenAI, 2024), Claude (Anthropic, 2024), Deepseek-R1 (DeepSeek-AI et al., 2025) and Llama (Dubey et al., 2024), how to effectively harness their capabilities in workflows remains a significant challenge. Directly querying these powerful models often yields inadequate results on complex or domain-specific tasks. Meanwhile, fine-tuning strong models to achieve desired behaviors can be prohibitively expensive and even infeasible, especially with closed-source, commercial models. This raises a critical research question: how can we unleash the potential of powerful LLMs without directly finetuning them?

To this end, training-free methods have emerged as potential solutions, ranging from simple heuristics like Few-shot Prompting (Brown, 2020), Chain-of-Thought (COT) (Wei et al., 2022), In-context Vectors (Liu et al., 2024a) to more intricate hand-designed agentic workflows (Yao et al., 2023; Zhou et al., 2023; Zhong et al., 2024b; Lu et al., 2025b). While heuristic approaches enhance performance, they struggle with complex tasks requiring multi-step reasoning (Prasad et al., 2024). Sophisticated hand-designed workflows mitigate some

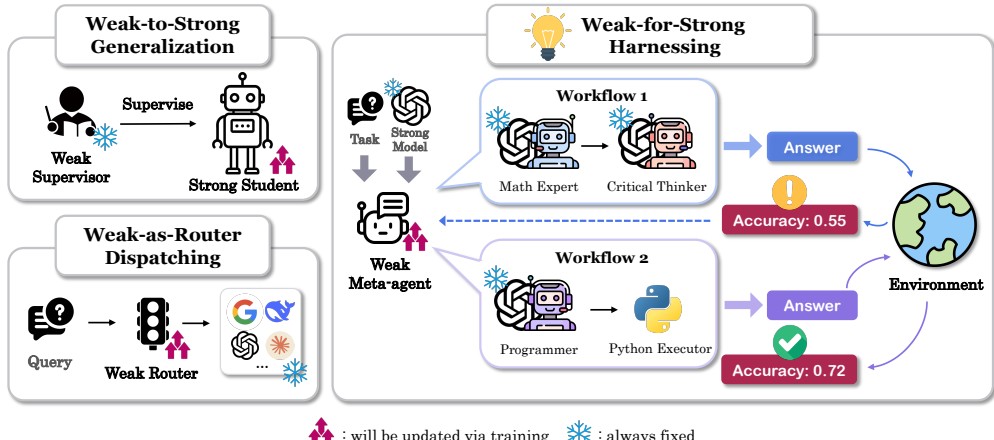

Figure 1: Comparison of paradigms: Weak-to-Strong Generalization uses weak models to supervise strong models, akin to superalignment; routing-based methods train weak models to dispatch queries across strong models; in contrast, Weak-for-Strong Harnessing (W4S) trains a weak model to optimize a strong model's performance on a specific task.

limitations but require labor-intensive trial-and-error and domain-specific manual tuning, resulting in high labor costs. Moreover, these manual strategies lack adaptability across tasks or models and fail to fully exploit LLM potential (Cemri et al., 2025), aligning with the "bitter lesson" (Sutton, 2019) that hand-engineered solutions are outpaced by adaptive, data-driven systems. Recent efforts have explored representing workflows as executable code, enabling powerful models like GPT-4o or Claude to automate workflow generation and optimization (Hu et al., 2024; Zhang et al., 2024a). However, these training-free approaches underutilize historical data and environmental feedback, sometimes performing no better than random workflow sampling (App. E.1), highlighting the inadequacy of such approaches in practice.

The challenge becomes even more pronounced with superintelligent models whose behaviors might not be fully predictable or comprehensible to human users (Burns et al., 2024), raising critical questions about the optimal strategies for their utilization. Given the limitations of existing training-free methods and the intractability of fine-tuning strong LLMs directly, this paper turns into the idea of training a weaker model that can understand the behaviors of strong models as well as the downstream task, to harness the strong models based on its understanding in the place of human.

**Our Contributions.** We propose a new paradigm: **Weak-for-Strong Harnessing** (W4S), which trains a weak model to leverage the strengths of strong models. W4S casts the problem of harnessing strong models as a workflow optimization problem, and employs a weak model as a meta-agent trained specifically for the problem. Unlike previous methods (Zhuge et al., 2024; Zhang et al., 2024a) that predefine agentic modules, we maximize the degree of freedom of the meta-agent by constraining only the workflow interfaces. This allows the meta-agent to design every internal component in freedom, including prompts, hyperparameters, and building blocks, enabling more expressive and tailored solutions. We formulate this as a multi-turn Markov decision process (MDP), and introduce *reinforcement learning for agentic workflow optimization* (RLAO) to teach the meta-agent to design and refine workflows. Through iterative interaction with both the task environment and the behavior of strong models, the weak meta-agent learns to design and improve workflows for strong models based on history and feedback.

Our approach introduces a novel perspective on the potential ways of interaction between weak and strong models, distinct from existing paradigms such as weak-to-strong generalization (Burns et al., 2024) and weak-dispatch-strong routing framework (Frick et al., 2025), as illustrated in Figure 1. This new paradigm emphasizes the weak meta-agent's role in unlocking latent capabilities of existing models without modifying them directly. Our paradigm is significantly more efficient and less expensive than finetuning strong

models directly, while outperforming both finetuning weak models on targeted tasks and training-free methods.

We conduct comprehensive evaluations across eleven widely adopted benchmarks, including question answering, mathematics, and code generation tasks. Empirical results demonstrate that a 7B meta-agent, trained with only one GPU hour on five tasks, can design workflows that effectively leverage strong models, significantly outperforming all the baselines. W4S surpasses manually designed methods by 3.3% $\sim$ 27.1% and outperforms the strongest automated design baseline by 2.9% $\sim$ 24.6%. Notably, the workflows generated by our method exhibit strong generalization and transferability across tasks and strong models, demonstrating the robustness and adaptability of the learned weak meta-agent in orchestrating high-performance workflows.

## 2 Method: Weak-for-Strong Harnessing

This section presents the Weak-for-Strong Harnessing (W4S) framework that trains weak models to optimize agentic workflows for stronger models. The key insight is that workflow optimization can be formulated as a sequential decision-making problem where a weak meta-agent iteratively improves workflows through interactions with an environment, guided by performance feedback.

Specifically, we define an agentic workflow $W$ as a structured and executable Python function that internally invokes a strong model to perform specific downstream tasks. The W4S framework operates as an iterative process of workflow generation, execution, and refinement, as depicted in Figure 2(a), and is unfolded as follows:

- **Workflow Generation.** The weak meta-agent analyzes the task, historical workflows, and prior feedback to design a new workflow to leverage the given strong model, represented as executable Python code. A self-correction mechanism addresses coding errors.

- **Execution and Feedback.** The generated workflow is executed by a strong model on validation samples, producing performance feedback (e.g., Accuracy, Error Cases).

- **Refinement.** The meta-agent uses feedback to iteratively improve the workflow, adapting to the task and the strong model's behavior over multiple turns.

This process enables the meta-agent to learn task-specific strategies and harness the strong model's capabilities efficiently, without requiring direct fine-tuning of the strong model. To rigorously analyze this optimization problem, below we formalize it as a multi-turn Markov Decision Process (MDP), and present our Reinforcement Learning for Agentic Workflow Optimization (RLAO) algorithm for training the weak meta-agent.

### 2.1 Workflow Optimization as Multi-Turn MDP

An MDP is denoted by a tuple $\mathcal{M} = (\mathcal{S}, \mathcal{A}, \mathcal{P}, \mathcal{R})$, where $\mathcal{S}$ and $\mathcal{A}$ are the state space and the action space, respectively. In our case, $\mathcal{S}$ represents the current knowledge about the task, the model and workflow history, $\mathcal{A}$ consists of possible workflow designs, $\mathcal{P} : \mathcal{S} \times \mathcal{A} \times \mathcal{S} \to [0, 1]$ is the transition probability function, and $\mathcal{R} : \mathcal{S} \times \mathcal{A} \times \mathcal{S} \to \mathbb{R}$ is the reward function.

For each iteration $i$, the agent takes action $a_i$ at state $s_i$ according to a learnable policy $\pi_\theta(a|s) : \mathcal{S} \times \mathcal{A} \to [0, 1]$, where $\theta$ is the parameters of the meta-agent. The environment executes the workflow and provides feedback $f_i$ and feedback-based reward $r_i$, transiting to the next state $s_{i+1} = [s_i; a_i; f_i]$. This process continues for a fixed number of iterations or until a predefined convergence criterion is met, allowing the agent to refine workflows based on feedback.

**Initial State Setup.** The initial state $s_1$ consists of Instructions $\mathcal{I}$, Task description $\mathcal{T}$, Example workflow $w_0$ and its feedback $f_0$ (if available). Details about $\mathcal{T}$ and $w_0$ are shown in Appendix A.

$$s_1 = [\mathcal{I}; \mathcal{T}; W_0; f_0].$$

**Action Design.** Each action includes two steps: analysis and workflow generation.

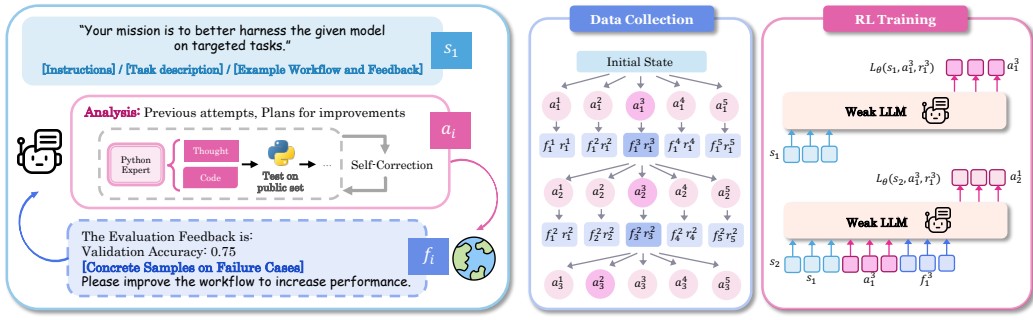

(a) W4S Optimization        (b) Overview of RLAO

Figure 2: (a) The weak meta-agent harness strong models by optimizing the workflows iteratively based on task and environment feedback. (b) To collect effective data for offline RL training, the meta-agent will sample $m$ times in each iteration, and using the best samples to form the next state. The data form multi-turn trajectories for offline RL training.

1. **Analysis**: The meta-agent is required to first conduct analysis include interpreting the task, history workflows, and feedbacks, and plans for improvements. Adding the analysis into the action space can bridge the gap between the pretrained language priors of LLMs and the environment, providing context for what adjustments should be made next.

2. **Workflow Generation**: Based on the analysis, the meta-agent produces function-represented workflow $W_i$. Unlike previous work such as Zhang et al. (2024a) that specifies predefined agentic modules (e.g., ensemble module, revision module), which constrains the creativity of LLMs, our approach only specifies the interface of the workflow function and provides helper functions like LLM calls and code execution. More details about the helper functions can be seen in Appendix A.2. This gives the meta-agent complete freedom to design the prompts, hyperparameters and internal logic of the workflow, fostering greater innovation and adaptability.

**Error Handling via Self-Correction.** To address potential coding errors in the generated workflows, we implement a self-correction mechanism by executing the workflow $W_i$ on a single validation sample. If execution fails due to bugs, the meta-agent will be prompted to perform self-correction to fix the identified bugs. This process can iterate up to 3 times, with the error message provided to the meta-agent at each step:

$$W_i^{(j+1)} = \text{SelfCorrect}(\text{Instructions}, W_i^{(j)}, \text{Error}_j).$$

where $W_i^{(j)}$ is the workflow at the $j$-th correction attempt and $\text{Error}_j$ is the corresponding error message. After self-correction, the complete action is then denoted as:

$$a_i = [\text{Analysis}_i; W_i].$$

where $W_i$ is now the workflow of the last correction attempt. If the workflow continues to produce errors after 3 correction attempts, the current iteration is skipped, and the erroneous workflow is not recorded.

**Evaluation Feedback.** Upon successful execution, the workflow is evaluated on both private and public validation sets to generate feedback:

1. Validation performance $v_i$: Accuracy measured on the private validation set.

2. Case studies: Examples of incorrect predictions from the public validation set, including input prompts, model answers, and correct answers.

The feedback is formally represented as:

$$f_i = [v_i; \text{CaseStudies}_i].$$

## 2.2 RLAO: Reinforcement Learning for Agentic Workflow Optimization

To train the weak meta-agent, we propose Reinforcement Learning for Agentic Workflow Optimization (RLAO), an offline RL algorithm tailored for this MDP, as shown in Figure 2(b). Online RL is less efficient due to the high cost of real-time workflow execution, so we collect trajectories offline and optimize the policy accordingly.

**Reward Mechanism.** Based on the feedback $f_i$, we define a reward $r_i$ as follows:

$$r_i = \begin{cases} 1, & \text{if } v_i > \max_{k \in [0, i-1]} v_k \\ 0.5, & \text{if } v_i > v_{i-1} \\ 0, & \text{otherwise} \end{cases}.$$

This reward function encourages both absolute improvement (surpassing all previous iterations) and relative improvement (surpassing the most recent iteration).

**Data Collection.** We collect a dataset of optimization trajectories for training the weak meta-agent. At each iteration $i$, we sample $m$ candidate actions. Subsequently, we select the best action based on validation performance to serve as the current action respectively to form the new state and execute the next action. Our dataset consists of both selected actions and unselected alternatives. At each iteration $i$, we generate $m$ candidate actions:

$$\{a_i^1, a_i^2, \ldots, a_i^m\}.$$

Then we select the best action based on validation performance:

$$a_i = a_i^* = \arg \max_{k \in [1, m]} v_i^k.$$

where $v_i^k$ represents the validation performance of the workflow produced by action $a_i^k$. This selection mechanism serves a dual purpose: it ensures that only the most effective workflow proceeds to the next iteration while simultaneously enriching our training dataset with both successful and unsuccessful attempts. This best-of-m approach helps to create high-quality trajectories for training while maintaining diversity.

**Policy Optimization.** We train the meta-agent using the offline variant of reward-weighted regression (RWR) (Lee et al., 2023; Qu et al., 2024) that optimizes the policy $\pi_\theta$.

$$\max_\theta \mathbb{E}_{\rho \sim \mathcal{D}} \left[ \sum_{t=1}^{T} \log \pi_\theta(a_t \mid s_t) \cdot \exp\left(\frac{r_t}{\tau}\right) \right] \tag{1}$$

where $\rho = (s_1, a_1, r_1, \ldots, s_T, a_T, r_T)$ is a trajectory from dataset $\mathcal{D}$, $T$ is the trajectory length, and $\tau$ is a temperature hyperparameter controlling reward scaling.

## 3 Experiments

### 3.1 Experimental Setup

**Baselines.** We compare workflows discovered by W4S against manually designed methods for LLMs, including 5-shot prompting, COT (Wei et al., 2022), Self Consistency CoT (5 answers) (Wang et al., 2022), Self-Refine (max 3 iteration rounds) (Madaan et al., 2023), LLM Debate (Du et al., 2023), Quality Diversity (Lu et al., 2025a) and Dynamic Assignment (Xu et al., 2023a). We also compare against workflow designed by automated workflow optimization method ADAS (Hu et al., 2024) and AFlow (Zhang et al., 2024a). Besides, we compare against a training-based baseline where GPT-4o-mini is fine-tuned on the validation dataset for fair comparison. More details are provided in Appendix D.2.

**Datasets.** We utilize eleven public benchmarks for our experiments: **(1) math reasoning**, we use MGSM (Shi et al., 2023), GSM8K (Cobbe et al., 2021), GSM Plus (Li et al., 2024a), GSM Hard (Gao et al., 2023), SVAMP (Patel et al., 2021) and MATH (Hendrycks et al., 2021). For

the MATH dataset, we follow (Hong et al., 2024a) in selecting 617 problems from four typical problem types (Combinatorics & Probability, Number Theory, Pre-algebra, Pre-calculus) at difficulty level 5; **(2) question-answering**, we use DROP (Dua et al., 2019) for evaluating reading comprehension, MMLU Pro (Wang et al., 2024) for evaluating multi-task problem solving and GPQA (Rein et al., 2023) for evaluating the capability of solving graduate-level Science questions; **(3) code generation**, we use HumanEval (Chen et al., 2021), and MBPP (Austin et al., 2021). For ADAS and AFlow, we conduct the searching for workflows on a validation set. For W4S, we further randomly split the validation set into a private validation set and a public validation set. All the evaluation results are conducted on the same held-out testing set. We follow the data splits used in established practices (Hu et al., 2024; Zhang et al., 2024a). More details about datasets can be found in Appendix. D.1.

**Metrics.** For HumanEval and MBPP, we report the pass@1 metric as presented in (Chen et al., 2021) to assess code accuracy. For multiple-choice datasets MMLU Pro and GPQA and mathematical datasets, we use Accuracy. For DROP, we report the F1 Score.

**Data Collection Details.** To manage computational constraints during training, we impose a trajectory truncation strategy in RLAO. Trajectories are limited to a horizon of $T = 2$ turns, with states reset every two iterations as follows:

$$s_{2i+1} = \begin{cases} s_1, & \text{if } i = 0, \\ [s_1; W_{2i}; f_{2i}], & \text{if } i > 0, \end{cases} , \quad s_{2i+2} = [s_{2i+1}, a_{2i+1}, f_{2i+1}].$$

where $s_1$ is the initial state, $W_{2i}$ is the workflow from the previous selected action, and $f_{2i}$ is its feedback. This results in a dataset $\mathcal{D}$ comprising single-turn trajectories (from unselected actions) and two-turn trajectories (from selected actions), formally:

$$\mathcal{D} := \left\{ \left( s_t^j, a_t^j, f_t^j, r_t^j \right)_{t=1}^{T'} \right\}_{j=1}^{|D|}, \quad T' \in \{1, 2\}.$$

For the following experiments results, we set $m = 5$ candidate actions per iteration to collect offline data, yielding 212 trajectories for Table 1 and 145 trajectories for Table 2.

**Implementation Details.** For ADAS and AFlow, we use `GPT-4o` as the meta-agent. For W4S, we employ and train `Qwen2.5-Coder-7B-Instruct` as the weak meta-agent. We also report the performance of directly utilizing `GPT-4o` without RLAO as meta-agent or training meta-agent with SFT on our framework in ablation studies. For execution, we employ `GPT-3.5-Turbo`, `GPT-4o-mini` in main text. More experiments using `GPT-4o` and `Claude-Sonnet` as executors are shown in Appendix E. We set iteration rounds to 20 for AFlow, and 30 for ADAS, following their original settings. We set iteration rounds to 10 for W4S. Training is conducted on 2 Nvidia H100 GPUs with a learning rate of 1e-5. The temperature $\tau$ for weighting the reward is set to 0.4. At inference time, W4S only samples one action in each iteration. More implementation details can be seen in Appendix D.

## 3.2 Experimental Results

**W4S significantly outperforms baseline methods across seen and unseen tasks.** As illustrated in Table 1, W4S, employing a 7B model as a weak meta-agent trained with RLAO, markedly surpasses few-shot learning, manually designed workflows, and automated workflow baselines with only 10 iterations. In this experiment, the meta-agent is trained on five tasks (DROP, MMLU Pro, MBPP, GSM Hard, Math) and generalize to two unseen tasks. The execution LLM is `GPT-4o-mini`. 'Finetuned GPT-4o-mini' represents using surpervised learning to train GPT-4o-mini on validation dataset, which yields unsatisfactory results, highlighting that leveraging a weak model trained via RLAO effectively outperforms direct fine-tuning on strong models under limited data conditions. Besides, 'W4S w/ SFT' represents training the weak model using the same data of RLAO with SFT. Notably, W4S with RLAO outperforms its untrained and SFT trained counterpart, further demonstrating the effectiveness of RLAO.

**W4S demonstrates generalization capabilities across different mathematical tasks.** Table 2 evaluates the generalization of W4S on mathematical reasoning tasks. Despite being

| Method | Seen Task | | | | | Unseen Task | |
|---|---|---|---|---|---|---|---|
| | DROP | MMLU Pro | MBPP | GSM Hard | Math | GPQA | HumanEval |
| 5-shot | 80.9 | 60.8 | 69.5 | 43.0 | 57.1 | 37.4 | 87.8 |
| Finetuned GPT-4o-mini | 75.9 | 61.1 | 76.2 | 41.2 | 56.8 | 41.8 | 82.8 |
| **Hand-designed Workflows** | | | | | | | |
| CoT | 78.5 | 56.6 | 72.4 | 39.5 | 56.9 | 36.7 | 88.8 |
| COT SC | 84.2 | 58.0 | 74.2 | 45.0 | 58.1 | 39.4 | 90.3 |
| Self Refine | 79.1 | 57.5 | 70.4 | 47.5 | 53.0 | 38.4 | 85.0 |
| LLM Debate | 83.0 | 60.1 | 73.9 | 49.5 | 53.9 | 40.8 | 89.1 |
| Quality Diversity | 80.0 | 59.1 | 71.8 | 46.5 | 55.3 | 40.1 | 86.0 |
| Dynamic Assignment | 80.2 | 57.4 | 71.8 | 41.5 | 56.9 | 36.0 | 90.1 |
| **Training-free Automated-designed Workflows** | | | | | | | |
| ADAS (30iter) | 82.0 | 58.4 | 74.0 | 52.5 | 51.4 | 39.6 | 90.8 |
| AFlow (20iter) | 80.6 | 59.2 | 83.9 | 52.0 | 58.4 | 42.0 | 92.1 |
| W4S w/o RLAO (10iter) | 85.3 | 61.0 | 86.0 | 60.6 | 58.6 | 39.8 | 92.7 |
| W4S w/ SFT (10iter) | 85.2 | 63.0 | 72.4 | 57.2 | 61.9 | 39.6 | 94.3 |
| **W4S (10iter)** | **87.5** | **64.8** | **86.8** | **76.6** | **63.0** | **45.9** | **95.4** |

Table 1: Comparison of performance (%) between W4S and baselines. All methods are executed using `GPT-4o-mini`, with each tested three times, and average results reported.

| Method | Seen Task | | Unseen Task | | |
|---|---|---|---|---|---|
| | GSM Plus | MGSM | GSM8k | GSM Hard | SVAMP |
| **Hand-designed Workflows** | | | | | |
| CoT | 24.5 | 28.0 | 38.5 | 14.0 | 77.8 |
| CoT SC | 27.1 | 28.2 | 43.0 | 15.0 | 78.2 |
| Self Refine | 25.8 | 27.5 | 40.5 | 14.5 | 78.5 |
| LLM Debate | 29.9 | 39.0 | 49.0 | 18.0 | 76.0 |
| Quality Diversity | 21.1 | 31.1 | 29.0 | 14.0 | 69.8 |
| Dynamic Assignment | 27.1 | 30.1 | 34.0 | 19.5 | 73.0 |
| **Training-free Automated-designed Workflows** | | | | | |
| ADAS (GPT-4o 15iter) | 52.0 | 47.5 | 54.5 | 31.5 | 80.8 |
| ADAS (GPT-4o 30iter) | 57.4 | 53.4 | 61.1 | 34.5 | 82.8 |
| AFlow (GPT-4o 20iter) | 62.8 | 54.8 | 76.8 | 40.6 | 81.3 |
| **In-distribution Domains** | | | **Generalize to Other Math Domains** | | |
| **W4S (10iter)** | **68.2** | **66.2** | **86.5** | **61.8** | **84.2** |

Table 2: Comparison of performance (%) between W4S and baselines. All methods are executed using `GPT-3.5-Turbo`, with each tested three times, and average results reported.

trained solely on GSM Plus and MGSM, W4S achieves substantial improvements over all baselines when tested on unseen tasks such as GSM8K, GSM Hard, and SVAMP. Particularly, W4S exceeds the strongest baseline methods by 10% on GSM8K and 20% on GSM Hard, highlighting W4S as a scalable and effective method for harnessing powerful executors.

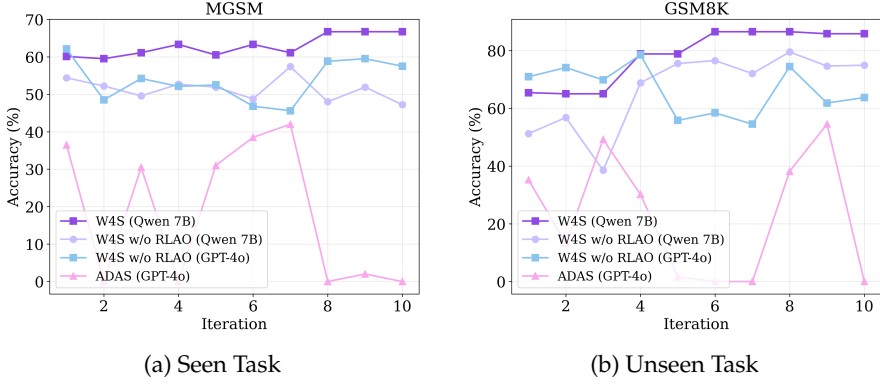

(a) Seen Task                    (b) Unseen Task

Figure 3: Ablation Studies on MGSM and GSM8K. The purple line represents the performance of W4S using 7B model trained on MGSM and GSM Plus with RLAO.

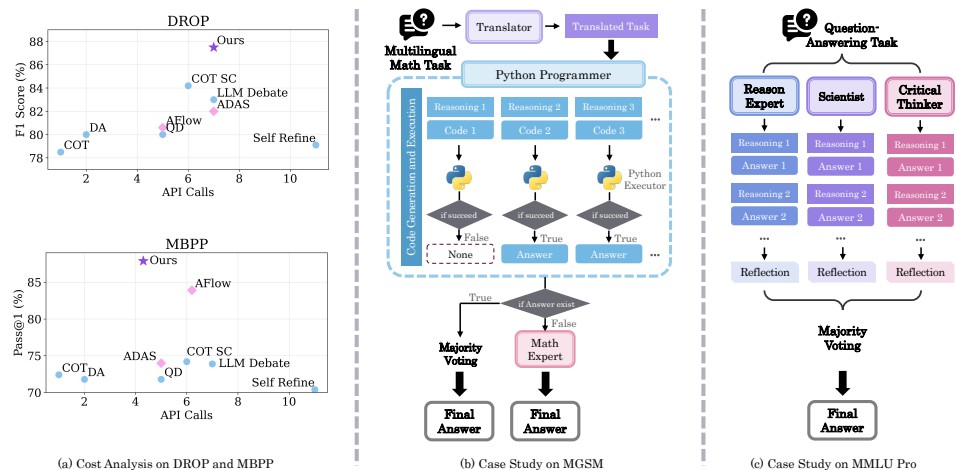

Figure 4: Cost Analysis (a) and Case Studies (b, c) of W4S on different benchmarks.

| Method | Workflow Optimization | | | Execution on Testing Set | | | |
|---|---|---|---|---|---|---|---|
| | Wall-clock Time (min) | Meta-Agent Cost ($) | Execution Cost ($) | Wall-clock Time (min) | Inference Cost ($) | Total Cost ($) | Pass@1 |
| ADAS | 131 | 11.3 | 9.0 | 4.0 | 0.6 | 20.9 | 90.8 |
| AFlow | 61 | 0.6 | 0.4 | 10.9 | 0.3 | 1.3 | 92.1 |
| **W4S** | **33** | 0 | 0.4 | **2.7** | 0.5 | **0.9** | **95.4** |

Table 3: Efficiency comparison between W4S and state-of-the-art baselines on HumanEval, using `GPT-4o-mini` as the executor. Testing set execution metrics are averaged over three runs, with costs reported for all runs.

**Ablation Study.** Figure 3 illustrates iteration curves for MGSM (seen task) and GSM8K (unseen task). W4S, leveraging a weak meta-agent trained via RLAO, demonstrates stable and consistent improvements over iterations on both seen and unseen tasks. Conversely, ADAS, employing `GPT-4o` directly as the meta-agent, has very random performance and often output workflow with a performance of 0. Besides, W4S trained with RLAO outperforms directly using the 7B model without training, demonstrating the efficacy of our training method. Notably, utilizing trained weak meta-agent also outperforms directly using a strong model like `GPT-4o` to optimize the workflow, validating the necessity and effectiveness of our weak-for-strong paradigm facilitated by RLAO training.

**Cost Analysis.** In Figure 4(a), we demonstrate the comparison of performance and API calls between the baselines and the workflows found by ADAS, AFlow (using GPT-4o as meta-agent) and W4S (using trained 7B model as meta-agent) on DROP and MBPP, and using GPT-4o-mini as execution LLM. Results demonstrate that W4S can design workflows that harness strong models to have a better performance with less test-time compute compared with hand-designed workflows. Besides, by automating the design of effective agentic workflows, W4S eliminates the human labor costs previously required. Although W4S adds more cost of training, this training cost is negligible compared to finetuning a strong model on targeted task. Training a 7B model on five tasks in Table 1 requires only one GPU hour, which can actually be amortized over repeated use across different benchmarks. Table 3 provides a detailed efficiency comparison on an unseen benchmark, including API cost and wall-clock time and testing performance. Compared to ADAS and AFlow, W4S achieves a Pass@1 score of 95.4 with a significantly reduced optimization time (33 minutes) and zero meta-agent API cost. Test-time execution remains comparable to baselines, with a wall-clock time of 2.7 minutes and an inference cost of $0.5, underscoring W4S's ability to balance efficacy and efficiency.

**Case Study.** Figure 4(b) and (c) visualizes the workflows designed by W4S on MGSM and MMLU Pro. For MGSM, the workflow employs a Translator LLM that converts multilingual problems to English, followed by a Python Programmer generating multiple code implementations. Successful code executions are aggregated via Majority Voting, with a Math

Expert as fallback for challenging problems. This adaptive approach dynamically adjusts strategies based on execution results. For MMLU Pro, W4S creates a parallel multi-agent workflow with specialized experts that each develop multiple reasoning paths. After a Reflection phase where agents review their answers, a Majority Voting mechanism produces the final answer. Both workflows demonstrate how W4S automatically discovers task-specific decomposition strategies and effective coordination mechanisms that combine specialized expertise with critical evaluation.

## 4 Related Works

**Agentic Workflows.** Agentic workflows and autonomous agents represent distinct LLM application paradigms: the former follows structured, multi-step processes, while the latter dynamically solves problems. Unlike agents requiring custom decision patterns, agentic workflows leverage human expertise for automated construction. They have been applied to problem-solving (Wei et al., 2022; Wang et al., 2022; Madaan et al., 2023; Wang et al., 2023; Han et al., 2025; Zhou et al.), code generation (Hong et al., 2024b; Ridnik et al., 2024; Zhong et al., 2024a), data analysis (Xie et al., 2024; Ye et al., 2024; Zhong et al., 2024a; Zhou et al., 2023), and mathematics (Zhong et al., 2024b; Xu et al., 2023b).

Recent research automates workflow design via prompt tuning (Fernando et al., 2024; Yüksekgönül et al., 2024; Yang et al., 2024; Khattab et al., 2024; Liu et al., 2024b), hyperparameter optimization (Saad-Falcon et al., 2024), and end-to-end workflow optimization (Li et al., 2024b; Zhou et al., 2024a; Zhuge et al., 2024; Hu et al., 2024; Yin et al., 2024). Methods like GPTSwarm (Zhuge et al., 2024), ADAS (Hu et al., 2024) and AFlow (Zhang et al., 2024a) explore structured representations, yet efficient workflow discovery remains a challenge. Unlike previous methods relying on human-defined logic, our approach employs reinforcement learning (RL) to autonomously optimize workflows, achieving superior scalability and performance. Besides, unlike previous methods that treat workflows as graphs with predefined agentic modules as nodes, we maximize the creativity of the meta-agent by constraining only the workflow interfaces.

**Weak-to-Strong Generalization.** Weak-to-strong generalization refers to stronger models outperforming weaker supervisors after fine-tuning. While Burns et al. (2024) empirically demonstrated this effect, its limitations remain. Theoretical analyses (Charikar et al., 2024; Lang et al., 2024) and practical approaches—including LLM debates (Kenton et al., 2024), easy-to-strong generalization (Sun et al., 2024), small model search (Zhou et al., 2024c), hierarchical mixture of experts (Liu & Alahi, 2024), reliability-aware alignment (Guo & Yang, 2024), alignment with weak LLM feedback (Tao & Li, 2024)—have been explored. Unlike prior work focused on supervised improvements, we introduce a learning-based agentic optimization approach to harness strong models via weak models.

**Concurrent Work.** MaAS, ScoreFlow, and MAS-GPT (Zhang et al., 2025; Wang et al., 2025; Ye et al., 2025) also explore automatic workflow generation for LLM-based systems. MaAS (Zhang et al., 2025) optimizes distribution over multi-agent architectures. ScoreFlow (Wang et al., 2025) conducts evaluation-based preference optimization, yet lacks interaction-driven refinement. MAS-GPT (Ye et al., 2025) conducts supervised learning and lacks feedback adaptation. In contrast, W4S trains a weak agent via RL to iteratively optimize workflows with environment feedback, achieving adaptive strong model harnessing.

## 5 Discussion

**Safety Considerations.** Although it is highly unlikely that the meta-agent employed in our setting generate malicious behaviors, they might inadvertently produce unsafe outputs due to limitations in model alignment (Rokon et al., 2020; Chen et al., 2021). We mitigate this risk through containerized execution of all generated code within secure, isolated environments, automated detection of potentially unsafe code patterns and manual safety inspections.

In fact, our training methodology offers an advantage from a safety perspective compared with training-free methods that rely directly on potentially less-aligned strong models to

design workflows. The weak meta-agent could be specifically trained to avoid generating workflows that might misuse the strong model's capabilities or produce harmful outputs. While we didn't explicitly optimize for safety in this paper, future work could integrate safety-oriented objectives by penalizing harmful patterns and rewarding safe workflows.

**Limitations.** The strong models we utilize are certainly powerful, but they do not represent the frontier of closed-source models, such as OpenAI o1 (OpenAI et al., 2024) and Deepseek R1 (DeepSeek-AI et al., 2025). As models continue to advance in capability, the gap between weak models and strong executors may widen, introducing new challenges. Additionally, our experiments focuses primarily on question-answering and reasoning datasets, representing only a slice of potential applications. Complex tasks like long-horizon planning and real-world agentic tasks may require further methodological refinements. Nevertheless, despite these limitations, our current results remain highly encouraging. They demonstrate the viability and effectiveness of training weak models to better understand the behaviors and leverage the potential of stronger models, suggesting a promising direction for future research as AI systems continue to advance in capability. Our work represents an important proof of concept that will become increasingly valuable as the capability gap between accessible and cutting-edge models continues to widen.

# 6 Conclusion

We propose Weak-for-Strong Harnessing (W4S), a novel framework that trains a weak meta-agent to design and optimize agentic workflows, effectively harnessing the capabilities of stronger language models. By formulating workflow optimization as a multi-turn MDP and leveraging Reinforcement Learning for Agentic Workflow Optimization (RLAO), our approach enables a 7B model to harness state-of-the-art models, achieving significant performance gains across diverse benchmarks. A key benefit of Weak-for-Strong is that the meta-agent is a smaller model that's easier and cheaper to train with RL and also easier to control because it's open source. As LLMs continue to advance, W4S establishes a promising paradigm for efficiently unlocking their potential, paving the way for future exploration into adaptive, learning-driven agentic systems.

# Acknowledgments

We would like to thank Leitian Tao, Siwei Han, Shirley Wu, Yuxiao Qu, Zhenting Qi and the members of Zou group for helpful advice and discussions.

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

# A  Technical Details

## A.1  Prompt

We use the following prompts for the meta agent in W4S.

---

**System Prompt for the Meta Agent**

```
You are an AI agent system improvement expert specializing in LLM prompting techniques
    and state-of-the-art LLM agent architectures. Your mission is to evolve and
    optimize agentic systems through innovative prompts, strategies, and architectural
     patterns. Your core focus is on continuously enhancing system performance through
    :

1. Careful analysis of historical agentic systems and their performance feedback
2. Creative exploration of novel architectures and techniques
3. Systematic improvement by optimizing the agentic system code based on empirical
    results

You will carefully study evaluation feedback to extract actionable insights and
    identify promising directions for improvement. Think critically about what worked,
     what didn't, and why. Use this understanding to design targeted enhancements
    while maintaining system stability.

Your improvements should push boundaries through principled innovation - each
    iteration building upon proven successes while thoughtfully exploring new
    approaches. Draw inspiration broadly from LLM agent research and other relevant
    fields.
```

---

**Main Prompt for the Meta Agent**

```
### **Agentic System Interface**:
Function you should optimize: `workflow(agent, task: str) -> dict`
- Description: Solve the target task using current agent.
- Input: task (str) - The question/problem to be solved.
- Output: dict with mandatory "answer" key containing the solution; The value of "
    answer" should be converted to a string.
- Available API:
[APIs]

### Task Description
The task your designed agentic system should solve is:

[TASK]

### History Agentic Systems
Here is the archive of the history agentic systems and their evaluation feedback.
'system code' is the code of the solver function
'eval_feedback' includes performance metrics and randomly selected validation samples:

[HISTORY]

### Output Format
You MUST respond with:
1. Your analysis

2. A complete implementation of the workflow function in a Python code block,
    formatted EXACTLY as follows:
```python
def workflow(agent, task: str):
    \"""
    Fill in your code here. Any helper functions or import should be included in this
        function.
    \"""
    return return_dict
```
```

---

**Prompt for the Self Correction when a runtime error occurs.**

---

```
Error during evaluation:
[ERROR]

WARNING: DO NOT USE ANY TRY-EXCEPT BLOCKS IN YOUR SOLUTION.
Your task is to fix the root cause of the error, not to catch it.

Requirements:
1. Analyze the error message in detail
2. Explain the specific changes needed to fix the core issue
3. Provide a clean implementation that solves the problem directly
4. Do not include any error handling or try-except blocks

Please strictly follow the following output format:

[Your analysis here]

Code:
```python
def workflow(agent, task: str):
    \"""
    Fill in your code here.
    \"""
    return return_dict
```
```

---

## A.2   Helper Function

We implement the following APIs for meta-agent to use within the workflow. The helper function description will be added into the main prompt for the meta agent.

---

**Available APIs.**

---

```
+ `agent.call_json_format_llm(messages, temperature, num_of_response, agent_role,
    return_dict_keys, instructions)`: Call OpenAI APIs and return a list of dictionary
    format responses containing the keys specified in `return_dict_keys`.

+ `agent.call_llm(messages, temperature, num_of_response, agent_role, instructions)`:
    Call OpenAI APIs and return a list of text format responses.

+ `agent.execute_code(code)`: Execute the code and return the output. The code MUST
    contain a `solution` function. The output of `execute_code(code)` will be the
    return value of the `solution` function if the code is executed successfully or
    raise an exception.

+ `agent.extract_answer_str(response)`: Extract the numeric or LaTeX answer from the
    LLM response (str).

+ `agent.extract_code_block(response, entry_point='solution')`: Extract the code that
    contains `def <entry_point>` from the LLM response (str).

+ `agent.test_on_public_test(task, solution_code, entry_point, test_loop)`: Execute
    solution code on public test set, return `results` (dict), `results['result']` is
    `True` or `False`, `results['solution']` is the updated solution code, `results['
    feedback']` is the feedback:
```

---

## A.3   Task Description

A task description $\mathcal{T}$ briefly describes the task and the input and output format.

### A.4 Example Workflow

Example workflows anchor the format of the workflow, showing the correct function signature and how to utilize LLM call helper function. This helps avoid syntactic errors or LLM call errors. Besides, the example workflow will be executed on validation set before the first iteration to provide feedback.

Example workflow $W_0$ is chosen by domain: (1) **code generation**, we choose CoT; (2) **question-answering**, we choose CoT and Self Consistency CoT; (3) **math reasoning**, we choose CoT, Self Consistency CoT, and a CoT-with-code variant that executes the generated code in Python.

## B  Case Study

### B.1  Case Studies for W4S

**The workflow generated for MBPP**

```python
def workflow(agent, task: str, entry_point: str):
    instructions = "Requirements:\n1. Please explain your solution step by step.\n2.
        The answer MUST be a valid Python function.\n3. Use clear variable names and
        add comments for clarity."
    prompt = f"Your Task: \n{task}\nGenerate the complete function below with the
        function name equal to {entry_point}: "

    messages = [{"role": "user", "content": prompt}]
    response = agent.call_json_format_llm(
        messages=messages,
        temperature=0.3,
        num_of_response=3,
        agent_role="Python Programmer",
        return_dict_keys=["reasoning", "answer"],
        instructions=instructions.strip(),
    )

    return_dicts = response
    correct_solution = None

    for return_dict in return_dicts:
        solution_code = return_dict.get("answer", "")
        results = agent.test_on_public_test(task, solution_code, entry_point,
            test_loop=3)
        if results['result']:
            correct_solution = results['solution']
            break

    if correct_solution is None:
        # If no correct solution is found, take the first one
        correct_solution = return_dicts[0]['answer']

    return_dict = {
        "answer": str(correct_solution),
        "reasoning": return_dicts[0].get("reasoning", ""),
    }

    return return_dict
```

**The workflow generated for DROP**

```python
def workflow(agent, task: str):
    """
    Solve the target task using current agent. Use `agent.call_json_format_llm` to
        call OpenAI APIs.
    Fill in your code here. Any helper functions or import should be included in this
        function.
    """
```

```
    instructions = """Requirements:
1. Please explain step by step.
2. Please answer the question directly.
3. The answer MUST be a concise string.
4. If the problem asks for a number, provide it in precise float form (e.g., use 3
    instead of 'three', use 93.09 instead of 93).
5. Ensure a deep understanding of the context provided in the passage.
"""

    messages = [{"role": "user", "content": f"# Your Task:\n{task}"}]

    # Generate multiple solutions with different temperatures
    responses = agent.call_json_format_llm(
        messages=messages,
        temperature=0.7,
        num_of_response=5,  # Generate 5 different solutions
        agent_role="read comprehension expert",
        return_dict_keys=["reasoning", "answer"],
        instructions=instructions.strip(),
    )

    answers = []
    for response in responses:
        try:
            answer = str(response.get("answer", ""))
            answers.append(answer)
        except:
            continue

    # Ensemble prompt to select the most consistent answer
    ensemble_prompt = f"Given the task as follows: \n{task}\nSeveral solutions have
         been generated to address the given question. They are as
         follows:\n{answers}\nCarefully evaluate these solutions and identify the
         answer that appears most frequently. This consistency in answers is crucial
         for determining the most reliable solution."
    ensemble_messages = [{"role": "user", "content": ensemble_prompt}]
    ensemble_response = agent.call_json_format_llm(
        messages=ensemble_messages,
        temperature=0.3,
        num_of_response=1,
        agent_role="read comprehension expert",
        return_dict_keys=["reasoning", "answer"],
        instructions=instructions.strip(),
    )[0]

    return_dict = {
        "answer": ensemble_response["answer"],
    }

    return return_dict
```

## The workflow generated for GSMHard

```
def workflow(agent, task: str):
    programmer_instructions = """
    You should generate valid Python code to solve the math problem. Requirements:
    1. The code must define a solution() function and return only the final numerical
        answer.
    2. Use only basic arithmetic operation.
    3. Do not introduce a dead loop.
    4. Ensure the code handles all edge cases and returns a float.
    """
    messages = [{"role": "user", "content": f"Write Python code to solve this math
         problem. The code should follow the requirements. Problem: {task}"}]

    responses = agent.call_json_format_llm(
        messages=messages,
        temperature=0.3,
        num_of_response=5,  # Generate 5 different solutions
        agent_role="Python programmer",
        return_dict_keys=["reasoning", "code"],
        instructions=programmer_instructions.strip(),
    )
```

```python
    answers = []
    for response in responses:
        try:
                code = response.get("code", "")
                result = agent.execute_code(code)
                if isinstance(result, (int, float)):
                    answers.append(result)
        except Exception as e:
                continue

    if not answers:
        # Fallback to LLM reasoning if no valid code is generated
        math_expert_instructions = """
        Requirements:
        1. Please explain step by step.
        2. The answer MUST be a float.
        """
        messages = [{"role": "user", "content": f"# Your Task:\n{task}"}]
        response = agent.call_json_format_llm(
            messages=messages,
            temperature=0.8,
            num_of_response=1,
            agent_role="math expert",
            return_dict_keys=["reasoning", "answer"],
            instructions=math_expert_instructions.strip(),
        )
        return_dict = response[0]
        return_dict["answer"] = str(return_dict.get("answer", "0.0"))
        return return_dict

    # Use self-consistency to get the most common answer
    from collections import Counter
    answer_counts = Counter(answers)
    most_common_answer = answer_counts.most_common(1)[0][0]

    return_dict = {
        "answer": str(most_common_answer),
    }

    return return_dict
```

### The workflow generated for MATH

```python
def workflow(agent, task: str):
    # Define the instructions for the LLM
    instructions = """
    Requirements:
    1. Please explain step by step.
    2. The answer MUST be formatted correctly.
    3. If the task requires a numerical answer, provide it as a precise number or
        LaTeX expression.
    """

    # Call the LLM with the task and instructions
    messages = [{"role": "user", "content": f"# Your Task:\n{task}"}]
    response = agent.call_llm(
        messages=messages,
        temperature=0.7,
        num_of_response=5,
        agent_role="math expert",
        instructions=instructions.strip(),
    )

    # Extract answers from the responses
    answers = [agent.extract_answer_str(response) for response in response]

    # Count the frequency of each answer
    from collections import Counter
    answer_counts = Counter(answers)

    # Get the majority answer
    majority_answer, _ = answer_counts.most_common(1)[0]
```

```python
        # Format the answer correctly
        try:
            # Try to convert the answer to a float
            float_answer = float(majority_answer)
            if float_answer.is_integer():
                majority_answer = int(float_answer)
            else:
                majority_answer = float_answer
        except ValueError:
            # If not a number, keep it as is
            pass

        # Create the return dictionary
        return_dict = {"answer": majority_answer}
        return return_dict
```

**The workflow generated for MMLU Pro**

```python
def workflow(agent, task: str):
    from collections import Counter
    import random

    def get_initial_responses(task, agent_role):
        messages = [{"role": "user", "content": f"# Your Task:\n{task}"}]
        responses = agent.call_json_format_llm(
            messages=messages,
            temperature=0.7,
            num_of_response=5,
            agent_role=agent_role,
            return_dict_keys=["reasoning", "answer"],
            instructions="Requirements:\n1. Please explain step by step.\n2. The
                answer MUST be A or B or C or D or E or F or G or H or I or J."
        )
        return responses

    def refine_response(task, initial_response, agent_role):
        messages = [
            {"role": "user", "content": f"# Your Task:\n{task}"},
            {"role": "assistant", "content": f"Your initial solution:\nReasoning:
                {initial_response['reasoning']}\nAnswer:
                {initial_response['answer']}"}
        ]
        refined_response = agent.call_json_format_llm(
            messages=messages,
            temperature=0.3,
            num_of_response=1,
            agent_role=agent_role,
            return_dict_keys=["revised_reasoning", "revised_answer"],
            instructions="Requirements:\n1. Consider other experts' solutions
                carefully.\n2. Provide improved reasoning if needed.\n3. The
                revised_answer MUST be A or B or C or D or E or F or G or H or I or
                J."
        )[0]
        return refined_response

    def get_final_answer(refined_responses):
        answers = [response['revised_answer'] for response in refined_responses]
        answer_counts = Counter(answers)
        most_common_answer = answer_counts.most_common(1)[0][0]
        return most_common_answer

    # Dynamic role assignment based on task complexity
    agent_roles = ["Knowledge and Reasoning Expert", "Scientist", "Critical Thinker"]
    if len(task.split()) < 20:
        agent_roles = agent_roles[:2]  # Simplified task, use fewer roles

    # Initial responses
    initial_responses = []
    for role in agent_roles:
        initial_responses.extend(get_initial_responses(task, role))

    # Refine responses
```

| Domain | Dataset | #Validation | #Test |
|---|---|---|---|
| Math Reasoning | MGSM | 128 | 800 |
| | GSM Plus | 128 | 800 |
| | GSM Hard | 128 | 800 |
| | GSM8K | 128 | 800 |
| | SVAMP | 128 | 800 |
| | MATH | 119 | 486 |
| Code Generation | MBPP | 86 | 341 |
| | HumanEval | 33 | 131 |
| Question Answering | DROP | 128 | 800 |
| | MMLU Pro | 128 | 800 |
| | GPQA | 60 | 138 |

Table 4: Dataset Statistics.

```python
refined_responses = []
for response in initial_responses:
    refined_responses.append(refine_response(task, response,
        random.choice(agent_roles)))

# Get final answer
final_answer = get_final_answer(refined_responses)

return_dict = {
    "answer": final_answer
}

return return_dict
```

## C   More Related Work

**LLM Post-Training.** Modern LLMs undergo various post-training processes to enhance task-specific capabilities and align outputs with human preferences, including instruction tuning (Zhang et al., 2024b; Muennighoff et al., 2023; Feng et al., 2024; Qi et al., 2024), preference learning (Rafailov et al., 2024), and reinforcement learning (DeepSeek-AI et al., 2025; Zhou et al., 2024b). Our W4S framework is most closely related to multi-turn RL algorithms for LLMs. Qu et al. (2024) employed multi-turn RL to train language models in self-correction and self-improvement, while Zhou et al. (2024b) developed hierarchical multi-turn RL for training LLMs on complex interactive tasks. Unlike these approaches that directly enhance model capabilities, W4S trains a weak meta-agent to harness stronger models without modifying their parameters.

## D   More Implementation Details

### D.1   Datasets

We evaluate W4S on eleven datasets, including mathematical reasoning, question answering and code generation. For MATH, MBPP and HumanEval, we follow the data splits in Zhang et al. (2024a). For the other datasets, we follow Hu et al. (2024) and randomly split the dataset into validation and test splits. The dataset statistics are included in Table 4.

### D.2   Baselines

We evaluate W4S against several established methods, organized into three categories. First, we include standard LLM approaches: Vanilla (direct LLM invocation) and 5-shot prompting. Second, we compare against six hand-designed agentic workflows: (1) Chain-of-Thought (COT) (Wei et al., 2022), (2) Self-Consistency with Chain-of-Thought (COT-SC) (Wang et al., 2022), (3) Self-Refine (Madaan et al., 2023), (4) LLM Debate (Du et al., 2023), (5) Quality Diversity (Lu et al., 2025a), and (6) Dynamic Assignment (Xu et al., 2023a). Finally,

| Hyperparameters | Value |
|---|---|
| Learning Rate | 1e-5 |
| Training Epochs | 4 |
| Number of GPUs | 2 |
| LR Scheduler | cosine |
| Per Device Batch Size | 1 |
| Gradient Accumulation Steps | 16 |

Table 5: Hyperparameters for Training with W4S.

we benchmark against two recent automated workflow design methods: ADAS (Hu et al., 2024) and AFlow (Zhang et al., 2024a).

In COT, we prompt the LLM to think step by step before answering the question. In COT-SC, we sample $n = 5$ answers and then perform an ensemble by either a LLM query (QA, Code task) or a majority voting (Math task). For Self-Refine, we allow up to five refinement iterations, with an early stop if the critic deems the answer correct. In LLM-Debate, each debate module is assigned a unique role, such as Math Expert or Physics Expert, and the debate lasts for two rounds. In Quality-Diversity, we conduct three iterations to collect diverse answers based on previously proposed ones. In Role Assignment, we use one LLM query to first choose a role from a predefined set, and then use another LLM query to answer the question by acting within the chosen role.

For the hand-designed workflow implementations, we adopt the standardized versions from the ADAS framework to ensure fair comparison. For AFlow, we reproduce the results using their official codebase and implementation.

### D.3 Details for Data Collection

In our experiments, we set the number of candidate samples $m = 5$ and select the best-performing action to determine the next state. We filter out actions yielding workflows with extremely poor performance to ensure quality. Trajectories are collected over a maximum of 10 iterations per task in Table 1 and 15 iterations per task in Table 2. To manage computational efficiency, we apply trajectory truncation with a horizon of $T = 2$, resetting the state every two iterations and correspondingly resetting the maximum historical validation performance.

### D.4 Implementation Details

**Hyperparameters for Fine-Tuning with W4S.** For finetuning, we utilize the TRL (von Werra et al., 2020) codebase, but we customize the loss function and the dataset preprocessing. The base models are directly loaded from Hugging Face: Qwen2.5-Coder-7B-Instruct. The hyperparameters used for finetuning are specified in Table 5.

**Hyperparameters for Inference.** For inference, we employ the meta-agent with a temperature of 0.5 to sample once for each iteration, different from best-of-$m$ sampling during training. In order to keep consistent with the training data, we also apply trajectory truncation during inference, with a horizon $T = 2$.

## E More Experimental Results

### E.1 Limitation of Previous Work

Figure 5 illustrates a key limitation of ADAS. The 'Sequential' condition reflects its standard setup, where the history archive is updated each iteration, while 'Random' involves generating 30 independent workflow samples in the initial iteration. The results reveal that ADAS's sequential performance is comparable to random sampling, with its peak accuracy failing to surpass the best outcome from the 30 random samples. This suggests that ADAS struggles to leverage historical information effectively for iterative improvement.

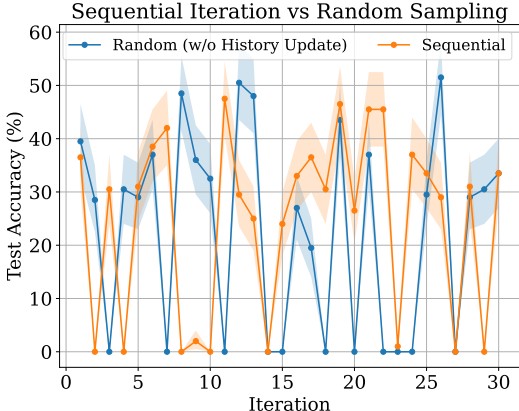

Figure 5: The Test Accuracy (%) of ADAS on MGSM dataset. 'Sequential' denotes the default configuration, updating the history archive iteratively; 'Random' indicates 30 independent workflow samples generated in the first iteration. Results show that ADAS's sequential performance closely mirrors random sampling, with its maximum accuracy not exceeding the best random sample.

| Execution LLM | GPT-4o | Claude-3-5-sonnet |
|---|---|---|
| Dataset | **MBPP** | |
| Vanilla | 75.9 | 77.7 |
| +W4S | 90.9 (+15.0%) | 89.8 (+12.1%) |
| Dataset | **GSM Hard** | |
| Vanilla | 55.0 | 53.8 |
| +W4S | 77.6 (+22.6%) | 78.2 (+24.4%) |

Table 6: Cross-model transferability of W4S. The meta-agent is trained for harnessing `GPT-4o-mini`. We report the performances before and after equipping the Execution LLM with the designed workflow.

## E.2 Cross-Model Transferability

Table 6 demonstrates the cross-model transferability of W4S. We train the meta-agent to optimize workflows for GPT-4o-mini, and directly transfer the workflow designed for GPT-4o-mini to other models.

## E.3 Cross-Dataset Transferability

Table 7 demonstrates the cross-dataset transferability of W4S. We train the meta-agent for GPT-4o-mini on one dataset, and directly transfer the optimal workflow to other datasets.

| Dataset | MBPP → H-Eval | GSM-Hard → MGSM | MMLU Pro → GPQA | GPQA → MMLU Pro |
|---|---|---|---|---|
| Vanilla | 87.7 | 82.9 | 39.1 | 56.1 |
| + W4S | 96.4 (+8.7%) | 87.4 (+4.5%) | 44.4 (+5.3%) | 64.1 (+8%) |

Table 7: Cross-dataset transferability of W4S. The Execusion LLM is GPT-4o-mini. "MBPP→H-Eval" means we train our meta-agent on MBPP, and evaluate on HumanEval. We report the performances before and after equipping the Execution LLM with the designed workflow.

### E.4 Comparison between Direct Task Training and W4S

We compare directly training a similarly sized open-weight model with GRPO on the validation dataset versus using W4S. On **GSM Hard**, W4S paired with a strong executor GPT-4o-mini remains substantially better under limited compute.

Table 8: Performance comparison between W4S with untrained baselines and directly training weak models on validation dataset.

| Model / Setting | Train? | Acc. |
|---|---|---|
| Qwen2.5-7B-Instruct (CoT) | No | 32.2 |
| Qwen2.5-7B-Instruct (GRPO) | Yes | 52.8 |
| GPT-4o-mini (CoT) | No | 39.5 |
| GPT-4o-mini (W4S) | Yes | **76.6** |

### E.5 Sensitivity to Initial Workflow $W_0$

We vary the initial example workflows $W_0$ and compare the performance after training. Table 10, 9 and 11 demonstrate the sensitivity results on three different datasets with different task types. 'CoT w/ Test' means using CoT strategy followed by testing the generated code on the public code test set. 'CoT w/ Code' means instead of instructing the models to generate text-based answer, instructing the models to generate step-by-step reasoning with a code as final output and executes the code to get the final answer.

Table 9: MBPP: sensitivity to $W_0$.

| $W_0$ | Acc. |
|---|---|
| CoT | 86.8 |
| CoT w/ Test | 87.6 |
| CoT + CoT-SC $\times$ 5 + CoT w/ Test | **87.9** |

Table 10: DROP: sensitivity to $W_0$.

| $W_0$ | Acc. |
|---|---|
| CoT | **87.9** |
| CoT-SC $\times$ 5 | 87.3 |
| CoT + CoT-SC | 87.5 |

### E.6 Helper Function Usage Statistics

We quantify the prevalence of tool/helper usage inside learned workflows.

### E.7 Training Cost Analysis

Training the weak meta-agent on five datasets (DROP, MMLU Pro, MBPP, GSM Hard, and Math) requires approximately 1 H100 GPU hour (30 minutes on 2 GPUs). Training on a single dataset requires only about 0.2 GPU hour. The API cost for collecting training trajectories varies by dataset, about 10\$ $\sim$ 20\$ USD per dataset, with GPT-4o-mini as executor LLMs. These computational and API cost could be further amortized when applying the trained meta-agent to multiple unseen datasets without additional training. We anticipate even stronger generalization capabilities when the meta-agent is trained across a more diverse range of domains.

Table 11: MGSM: sensitivity to $W_0$.

| $W_0$ | Acc. |
|---|---|
| CoT | 65.5 |
| CoT-SC $\times$ 5 | 66.8 |
| CoT w/ Code | 61.5 |
| CoT + CoT-SC + CoT w/ Code | **68.2** |

Table 12: Number of helper functions used per task (typical best workflows).

| | MATH | MBPP | MMLU-Pro | GSM Hard |
|---|---|---|---|---|
| # helper functions | 2 | 2 | 6 | 3 |

