# OpenReview forum: "Weak-for-Strong:  Training Weak Meta-Agent to Harness Strong Executors"
_colmweb.org/COLM/2025/Conference — COLM 2025_

### Official Review · Reviewer_vbtG · 2025-04-13

**Rating:** 6
**Confidence:** 4
**Ethics Flag:** 1

**Summary:**

This paper presents W4S, a meta-agent framework that aims to train a small model capable of designing agentic workflows that involve stronger models. The weaker model (a 7B Qwen instruct model) is trained to maximize the performance of a workflow on a validation set, optimized via offline RL with best-of-m rollouts. Using this meta-agent as a "controller" for stronger prompted models (GPT-3.5 and GPT-4),  authors show better workflows than hand-designed methods / existing optimization approaches across multiple benchmarks.

**Questions To Authors:**

- Typo: Figure 2 caption: using ⇒ use
- Equation below line 158 seems wrong. argmax is over the index in 1 ~ m, but the LHS assigns that to $a_i^*$.
- How does the best-of-m selection of action enriches training dataset with both successful and unsuccessful attempts?

**Reasons To Accept:**

- The motivation and overall pipeline of W4S framework seems very intuitive and promising. The framework reminds me of prompt-tuning methods such as textgrad, but this one focuses more on improving agent-style workflow which is timely.
- The authors provide results across multiple benchmarks and demonstrate consistent improvements in these tasks; notably the improvement does not require much compute despite involving additional training of the meta-agent.

**Reasons To Reject:**

- While I like the framing, "Weak-for-Strong" is misleading. In fact, off-the-shelf Qwen2.5-7B-Instruct (the weak model here) achieves much stronger performance in many tasks than what is reported as performance of W4S. For example, the weak model achieves 75.5 on MATH (https://qwenlm.github.io/blog/qwen2.5-llm/) while W4S with GPT-4o-mini (along with fine-tuned Qwen2.5-7B-Instruct as meta-agent) achieves 63.0 in Table 1.
- **Issues with clarity and (seemingly) arbitrary design choices**: A lot of components in the method but are not well motivated (at least does not explain why we need those components). For example, why do we need “example workflow” as input to the agent (line 115)? What exactly is the example workflow? How granular is the task T (i.e., does it require multiple calls to agents or tools, or is it just a one-shot API call to a stronger model)? How do you define "public and private validation sets" (line 139)? What happens when the current iteration is skipped due to error (line 143)? Do we have some fallback option? Some of these questions are not answered until the end of the paper, some gets only clearer after checking the examples in the appendix. Perhaps a better way of presentation is to include a concrete example in the method, and follow along that example for illustration.
- Adding to above, what is exactly done at test-time? The output of training phase would be the trained specialized meta-agent (based on Qwen2.5) capable of generating workflows. Now, at test-time, do we just sample a trajectory from the trained meta-agent? What is the input to the model during test time? Is it just the task description T (along with instruction, example workflow and feedback) for the benchmark of interest?
- **Need for Qualitative Analysis**: An important aspect of “workflow engineering” is that practitioners understand how qualitatively the generated workflows are different from canonical methods, as the proposed method introduces additional overheads such as model training and validation. The authors fail to deliver such qualitative analyses, such as how different the automated workflows are from typically used prompting methods, or what failure cases they bring in, or the impact of pre-defined helper functions (e.g., does the model learn to effectively use helper functions? What helped in what tasks and what did not?). In fact, the example workflow for GSM (line 595) doesn’t look any different from a typical Program-of-Thought prompting.
- As far as the reviewer knows, RWR is not an offline algorithm as authors state (https://arxiv.org/pdf/2107.09088, https://pure.mpg.de/rest/items/item_1790444_3/component/file_3075658/content). What the authors are doing is just a single iteration (in an EM framework) of RWR where the trajectories are sampled from current policy and then the policy is trained with return-weighted log likelihood.
- Overall, the tasks evaluated in the paper are relatively traditional problem-solving benchmarks such as GSM and DROP. While these benchmarks are easier to evaluate and iterate, they may not be the best place for sophisticated agentic workflows -- as noted in the first bullet point, off-the-shelf LMs are amazingly good at these tasks in zero-shot setting. I suggest the authors to consider more complicated benchmarks like SWEBench as the testbed for W4S, where simple CoT won't just work.

Overall, I think the suggested framework has a lot of potential, but the paper would benefit from another round of editing.

---

> ### Author Response · Authors · 2025-06-03
> **# Response to Reviewer vbtG (5/n)**
>
> > Choice and sophistication of evaluated tasks:
>
> * **Justification for chosen benchmarks and potential exploration of more complex tasks like SWEBench**:
>   **Answer**: Thanks for the suggestion. The tasks we use are following popular previous works (AFlow, ADAS), and other concurrent work (MaAS, ScoreFlow, MAS-GPT, etc). In order to test our framework on more difficult benchmark, we here provide experiments on AIME which is challenging even for the latest models.
>
> | Dataset (Executor)    | AIME (GPT-4o)  | AIME (GPT-4.1-mini) | AIME (o3-mini-medium) |
> | --- | ---- | ---- | ---- |
> | COT |  10.0  |  40.0 | 70.0 |
> | W4S | 23.3 | 70.0 | 90.0 |
>
> We thank the reviewer for pointing out SWE-bench. However, due to time limit, we cannot complete the experiments on SWE-bench during rebuttal. As mentioned in limitation part, we believe that extension to more complex agentic tasks like SWE-bench is a straightforward and interesting future step, which might involve some new improvements on the current framework like integrating more tools, memory, etc.
>
> ---
>
> > Additional Questions from Reviewer:
>
> * **Typo in Figure 2 caption**:
>   **Answer**: Thank you for pointing out the typo. We have corrected it in our paper.
>
> * **Correction for equation below line 158**:
>   **Answer**: Thanks for pointing out the issue. The equation should be $a_i^*
> \,=\,
> a_i^{\arg\max_{k \in [1,m]}
> v_i^k}$. We have corrected it in our paper.
>
> * **Explanation on how best-of-m selection enriches the training dataset**:
>   **Answer**: At every iteration the meta-agent draws m candidate workflows from its current policy and evaluates each on a held-out validation shard. We then log all m (state s, action a, reward r) triples but advance the search with only the highest-scoring candidate. This simple procedure enriches the offline buffer in two complementary ways:
>
>   * Built-in contrastive signal. Because the m actions share the same state, the buffer now contains one positive Δ-reward example alongside up to m – 1 neutral or negative siblings. The critic can therefore learn “this specific edit raised the strong model’s score, these nearly-identical edits did not,” which sharpens the advantage estimate far more than seeing isolated successes or a sea of uniformly low-return steps.
>   * Curriculum through winner continuation. Retaining only the best action to form the next state steers the roll-out toward promising regions of the workflow space. Subsequent m-tuples are sampled around progressively stronger baselines, so the agent observes a sequence of harder-to-improve contexts—an implicit curriculum that accelerates convergence and reduces variance.
>
>   Without best-of-m, a single draw per state would often yield an uninformative failure (no clue what would have worked) or a rare success with no contrasting negatives (no gradient direction).
>
>   Here we provide new experiments of W4S with best-of-m and without best-of-m (sample one action at each iteration). For the latter, we collect more trajectories to keep number of training data the same.
>
>
> Table: Performance of W4S on GPQA and GSM Hard. The Executor is GPT-4o-mini.
> |  | GPQA | GSM Hard |
> | -------- | -------- | -------- |
> | without best-of-m  | 40.1 | 65.0  |
> | with best-of-5    |  45.9  | 76.6  |
>
>
> ---
>
> We appreciate the reviewer's detailed feedback and believe these clarifications significantly strengthen the paper. We will include the new results and analysis in our paper.

---

> > ### Comment · Reviewer_vbtG · 2025-06-03
> >
> > Thanks for the detailed response. Most of my clarification questions are resolved, but I still have a fundamental concern whether the workflow we could obtain from W4S is going to qualitatively different from what we could get by manual validation. Perhaps (1) a more detailed comparison against competitive methods, and/or (2) application to agentic tasks (I feel like AIME is yet another example of problem-solving benchmark which is not fundamentally different from benchmarks used in the original version) where manual curation of workflow may be intractable, are necessary to mitigate this concern. But given that the author responses improved overall completeness and presentation, I will raise my score in the future iteration.

---

> > ### Author Response · Authors · 2025-06-09
> > **Response to Reviewer vbtG (4/n)**
> >
> > > New agentic dataset
> >
> > We evaluate W4S on GAIA-text (127 validation samples + 230 test samples) using gpt-4.1 as strong executor. GAIA is a widely used agentic benchmark that encompasses tasks from various domains such
> > as web browsing, file reading, calculation, etc. We additionally implement several tools including web search, PDF/image/video reader and python code generation/execution that strong executors can choose to use. In this benchmark, W4S learns to combine enhanced tool usage with iterative reflexion.
> >
> > Validation Accuracy on GAIA using GPT-4.1 as executor.
> > |      | Avg  | Level 1    | Level 2 | Level 3 |
> > | ---- | ---- | --- | --- | -------- |
> > | CoT  | 18.1 | 28.6 |   13.6  | 10.5  |
> > | CoT w/ Tool Use | 24.4 | 33.3 | 22.7 | 10.5 |
> > | CoT SC | 18.1 | 28.6 | 13.6 | 10.5 |
> > | Self-Refine | 21.3 | 28.6 | 19.7 |  10.5|
> > | LLM Debate|23.6| 35.7|18.2 |15.8 |
> > | ADAS | 22.0 | 18.2 | 31.0 | 15.8 |
> > | W4S  |  33.9   |   45.2  |   31.8  |  15.8   |

---

> > > ### Comment · Reviewer_vbtG · 2025-06-09
> > >
> > > Thanks for the detailed response. These additional results and analyses greatly improves the paper. I will increase my score assuming the revised version will reflect the new results and case studies.

---

> > > > ### Author Response · Authors · 2025-06-10
> > > > **Thanks for the response!**
> > > >
> > > > We sincerely thank the reviewer for the thoughtful reviews and for increasing the rating. We’re glad that our clarifications addressed the core concerns and will include the additional analysis into our paper.
> > > >
> > > > Thank you again for valuable suggestions that make our paper better!

---

> ### Author Response · Authors · 2025-06-03
> **# Response to Reviewer vbtG (4/n)**
>
> > Incorrect description of RWR:
>
> * **Clarification of the offline RL approach and single-iteration EM implementation**:
>   **Answer**: Thank you for pointing this out--indeed the original Reward Weighted Regression (RWR) was presented as an online EM-style: each iteration collects fresh trajectories and then performs a return-weighted maximum-likelihood (M-step) update.[1,2]. However, the RWR idea is not limited to online interaction and can be applied in an offline RL setting as well, either by collecting a single batch of trajectories to train a policy via one round of reward-weighted regression or by reusing the same offline datasets for multiple rounds of training. This “one-shot” variant has been widely adopted and explicitly described as offline or batch RL in prior work. For example, [3] re-uses a single batch of demonstrations for multiple RWR iterations, [4] describe a single-round reward-weighted fine-tuning of a diffusion model, which is essentially an offline RWR. [5-6] also collects one batch of trajectories for training LLMs.
> Our “Weak-for-Strong” setting matches these precedents: we collect one batch of trajectories (to avoid prohibitive API cost), weight them by the strong model’s reward signal, and perform a single supervised update. Empirically this already captures most of the potential gain, so further online iterations were unnecessary.
> We will revise the manuscript to read like: *"We employ the offline variant of Reward-Weighted Regression—a single M-step on a fixed dataset ([4-6])."*
>
> [1] Reinforcement Learning by Reward-weighted Regression for Operational Space Control
> [2] Reward-Weighted Regression Converges to a Global Optimum
> [3]Reward-weighted regression with sample reuse for direct policy search in reinforcement learning
> [4] Aligning text-to-image models using human feedback.
> [5] Recursive Introspection: Teaching Language Model Agents How to Self-Improve
> [6] Aligning Language Models with Offline Learning from Human Feedback

---

> ### Author Response · Authors · 2025-06-03
> **# Response to Reviewer vbtG (3/n)**
>
> > Need for Qualitative Analysis:
>
> Thanks for the suggestion to enhance our paper. Here we provide more qualitative analysis to further prove the effectiveness of our framework.
>
> - **Cost–benefit trade-off:** The extra training/validation overhead noted by the reviewer is modest compared with manual prompt engineering. Although our method, as well as other automated workflow design methods require additional training and validation, automated workflows are still cost-efficient since human labor is the most expensive. It takes time for human to try different prompting methods, come up with new approaches, refine prompts for specific tasks.
>
> - **How the learned workflows differ from canonical prompting:** Canonical techniques such as chain-of-thought or program-of-thought supply a single, static prompt; in contrast a W4S workflow is a program that decides, step by step, which tool (strong LLM call, Python, majority vote, fallback policy, translator, etc.) to invoke next, conditioned on previous outputs. For instance, on GSM-Hard the agent first generates a concise textual reasoning, then dispatches the resulting numerical expression to the Python interpreter for exact arithmetic, and finally wraps the answer with majority-voting. If code execution fails, it falls back to CoT with role-playing; on MGSM it inserts an initial translation stage that never appears in English-only datasets. These multi-tool pipelines emerge automatically and differ markedly from hand-written program-of-thought prompts, which typically mix reasoning and calculation inside a single LLM call.
>
> - **Discussion on failure cases:**
>   * **Validation over-fitting.** Early agents occasionally hard-coded answers for samples in case studies. Splitting validation into public/private subsets and rewarding improvement on the private set mitigates the issue.
>   * **Exploration collapse into resampling loops:** We discovered that occasionally Weak coders sometimes inflate “n-best” parameters instead of changing logic. Using best-of-m approaches along with utilizing models better at code can mitigate this issue. Besides, we set a hard limit on the inference time to avoid from too many API calls in one workflows. Future steps can consider improve rewards to reward more creative workflows.
>   * **Syntax/runtime errors:** Prior to RLAO, weak models are prone to make simple coding mistakes. Implementing a decent self-correction mechanism has almost solved this issue.
>
> - **Impact of helper functions**: The most useful helper fuctions are LLM call function and Python interpreter. For tasks like MATH, the agent learns to use answer extraction function to extract answers in the response and then performs majority voting or debating. The helper function can also be seen as tools for meta-agent. Users can develop more helper functions if more complex agentic tasks needs.
>
>   Here we provide statistics of the number of helper function used for different tasks. We observe that workflows generated for different tasks include at least two helper functions.
>
> |                         | MATH | MBPP | MMLU Pro    | GSM Hard |
> | ----------------------- | ---- | ---- | --- | -------- |
> | # helper function usage | 2    | 2    | 6  | 3        |

---

> ### Author Response · Authors · 2025-06-03
> **# Response to Reviewer vbtG (2/n)**
>
> > Clarity and Design Choices:
>
> We thank the reviewer for suggestions to enhance our paper clarity.
>
> * **Motivation and explanation for “example workflow” (line 115)**:
>
>   **Answer**: The “example workflow” serves two roles.
>
> First, it anchors the format of the workflow, showing the correct function signature and how to utilize LLM call helper function `agent.call_json_format_llm`. This helps avoid syntactic errors or LLM call errors. Here we show an example:
>
> ```python
> def workflow(agent, task: str):
>     instructions = "Requirements:\\n1. Please explain step by step."
>     messages = [{"role": "user", "content": f"# Your Task:\\n{task}"}]
>     response = agent.call_json_format_llm(
>         messages=messages,
>         temperature=0.5,
>         num_of_response=1,
>         agent_role="expert",
>         return_dict_keys=["reasoning", "answer"],
>         instructions=instructions.strip(),
>     )
>
>     return_dict = response[0]
>     return_dict["answer"] = str(return_dict.get("answer", ""))
>     return return_dict
> ```
> Second, the example workflow will be executed on validation set before the first iteration to provide feedback. Our rewards are based on current as well the history workflows. The example workflow enables the system to compute reward signals even before the meta‐agent has generated any workflows.
>
>
> * **Definition and granularity of task T**:
>   **Answer**: For the Task description $T$ included in meta-agent's state, it's text that briefly describe the task, the input and output format. We will list all $T$s to Appendix.
>
>   As for the specific task, it depends on different datasets. For our current QA benchmarks, the task will simply be the questions. For example, a math question is a task. But if we extend it to more complex agentic benchmark, a task can be a complex one that might need multiple turns, tool usages, etc.
>
> * **Clarification on public/private validation sets (line 139)**:
>   **Answer**: Thanks for pointing out. We will add more details for public/private sets in our paper. Public and private validation sets are created by randomly splitting the original validation data. During training, validation accuracy is computed only on the private split, while case studies (including prompts, model answers, ground-truths, etc) are drawn from the public split (10 validation samples). By exposing the meta‐agent to case studies only from public examples, we prevent it from overfitting to validation samples. In practice, we have observed instances where the meta‐agent attempts to generate workflows that simply reproduce known question‐answer pairs from public examples. Because such memorization does not improve accuracy on the private split, these behaviors receive low rewards and are thus discouraged.
>
> * **How iteration is skipped and fallback options (line 143)**:
>   **Answer**: During data collection we sample m candidate workflows. Any candidate that fails to execute is simply discarded. If the agent only samples one action at each iteration (without best-of-m) or all m candidates fail (we never see this happens though), the iteration will be skipped, which means we keep the same state to resample actions but the count of iteration will plus one. In fact, with our self-correction mechanism, these cases rarely happens in . And during testing time, with our trained models, we never witness such thing happens. We do not have other fallback options and do not inject incorrect workflows to the history because they will propogate errors.
>
> * **Clarification of test-time procedures**:
>   **Answer**: We provide details of test-time procedure in *Implementation Details* in Section 3.1.  Similar to ADAS and AFlow, W4S samples a trajectory at test time with only one action in each iteration. The difference is that ADAS and Aflow requires 30/20 iterations each while W4S only requires 10 iterations to converge. The input is just what you said, the task description, instructions, example workflow, feedback of the specific benchmark. We will add this information to Section 3.1.

---

> ### Author Response · Authors · 2025-06-03
> **# Response to Reviewer vbtG (1/n)**
>
> We thank the reviewer for the insightful feedback and valuable suggestions. Below, we address each raised point individually:
>
> ---
>
> > Misleading framing of "Weak-for-Strong":
>
> * **Clarification and justification of the terminology and reported performance differences**:
>   **Answer**: Thanks for the question. First we have to clarify that, for MATH, we only use questions with the highest difficulty level (5), while in Qwen's report, they report the number on the whole dataset. If evaluating on full dataset, GPT-4o-mini performs much better than Qwen2.5-7B-Instruct. In most of the tasks, GPT-4o-mini has a much better performance than Qwen-7B except for GSM8K, where they have comparable performance since Qwen has been post-trained on similar sources. We already have experiments using GPT-4o and Claude-3-5-sonnet as executors in Appendix E.2, which are definitely strong models compared with 7B Qwen. We here also provide more experiments using GPT-4o, GPT-4.1-mini and o3-mini-medium as executors on AIME dataset.
>
>
> | Executors | GPT-4o | GPT-4.1-mini | o3-mini-medium |
> | -------- | -------- | --- | -------- |
> |  COT   |   10.0  |  40.0 |  70.0 |
> | W4S   | 23.3  |  70.0 | 90.0    |

---

> ### Author Response · Authors · 2025-06-09
> **Response to Reviewer vbtG (1/n)**
>
> We thank the reviewer for the insightful comments and are pleased that most of your concerns are addressed. Below, we provide (1) a detailed comparison of workflows produced by W4S against competitive methods (ADAS and AFlow), and (2) extend our analysis to a realistic agentic task scenario using the GAIA benchmark.
>
> ---
>
> > comparison with competitive methods
>
> We compare the workflows generated by W4S, ADAS, and AFlow on MBPP (coding task) and DROP (reasoning task):
>
>
> |  | ADAS | AFlow | W4S | Cost / Performance outcome |
> | -------- | -------- | --- | --- | -------- |
> | MBPP  |  Four role-specialized agents propose solutions, ranked and refined by feedback agents in multiple stages.  |  Generate five solutions, ensemble to select one, validate on public tests, refine if failed, iterate twice if needed. |   Generate three solutions, immediately validate via public tests, iterate twice if needed, return first valid solution.  | Performance: W4S > AFlow > ADAS; Cost: W4S < AFlow < ADAS|
> | DROP | CoT SC (provided initially) | Multiple specialized roles iteratively refine solutions. | CoT ×5 with detailed ensemble instructions and optimized temperatures. | Performance: W4S > ADAS > AFlow  |
>
> On DROP, ADAS generates 30 intricate workflows for 30 iterations that still underperform compared to simple CoT SC. For our W4S, the generated workflow is also similar to CoT SC, but with a more detailed and instructive prompts and suitable temperatures. Since W4S utilizes best-of-m selection and conducts RL finetuning, it learns that for this task, combining resampling with ensemble or even simply majority voting is better than other complex structures, it also learns to optimize prompts and temperatures.
>
> Besides, we want to highlight that  automated workflow generation via these different methods eliminates expensive manual validation. Although manually created workflows might appear similar, the process to achieve such optimized hyperparameters, prompts, and combination of techniques manually is highly time-consuming and costly. Furthermore, W4S often uncovers novel and task-specific combinations of existing methods that are non-trivial to discover manually.
>
> + MBPP coding task
>
>  Workflow generated by ADAS
>  ```python
> def forward(self, taskInfo):
>     # Step 1: Initial Competitive Stage
>     expert_instruction = "Please independently think step by step and provide a solution based on your expertise."
>     expert_roles = ["Software Engineer", "Data Scientist", "Python Developer", "Algorithm Specialist"]
>     expert_agents = [LLMAgentBase(["thinking", "answer"], "Expert Agent", role=role) for role in expert_roles]
>
>     # Gather initial solutions
>     initial_solutions = [agent([taskInfo], expert_instruction) for agent in expert_agents]
>
>     # Step 2: Role Adjustment and Feedback Structuring
>     feedback_instruction = "Evaluate the effectiveness of the provided solution and rank its usefulness."
>     feedback_agent = LLMAgentBase(["feedback"], "Feedback Agent")
>
>     feedbacks = [feedback_agent([taskInfo, solution[0], solution[1]], feedback_instruction)[0] for solution in initial_solutions]
>     ranked_solutions = sorted(zip(feedbacks, initial_solutions), key=lambda x: float(x[0].content) if x[0].content.isdigit() else 0, reverse=True)
>
>     # Adjust roles: prioritize collaboration with top solutions
>     top_solutions = [solution for _, solution in ranked_solutions[:2]]
>     role_adjustment_instruction = "Based on ranking, guide collaboration to refine top solutions."
>     refined_solutions = []
>     for solution in top_solutions:
>         refined_solution = LLMAgentBase(["thinking", "answer"], "Refinement Agent")([taskInfo] + solution, role_adjustment_instruction)
>         refined_solutions.append(refined_solution)
>
>     # Step 3: Collaborative Refinement Stage
>     collaboration_instruction = "Collaborate and integrate the refined insights into a cohesive solution."
>     collaboration_agent = LLMAgentBase(["thinking", "answer"], "Collaboration Agent")
>     collaboration_thinking, collaboration_answer = collaboration_agent([taskInfo] + [info for solution in refined_solutions for info in solution], collaboration_instruction)
>
>     # Step 4: Final Decision
>     final_decision_instruction = "Converge the collaborative insights into a final coherent answer."
>     final_decision_agent = LLMAgentBase(["thinking", "answer"], "Final Decision Agent", temperature=0.1)
>     thinking, answer = final_decision_agent([taskInfo, collaboration_thinking, collaboration_answer], final_decision_instruction)
>     return answer
>  ```

---

> > ### Author Response · Authors · 2025-06-09
> > **Response to Reviewer vbtG (2/n)**
> >
> > Workflow generated by AFlow (MBPP)
> >  ```python
> > FEEDBACK_SUGGESTIONS = "Analyze the current code solution, and suggest improvements specifically targeted towards solving core issues or enhancing performance and reliability."
> > class Workflow:
> >     def __init__(
> >         self,
> >         name: str,
> >         llm_config,
> >         dataset: DatasetType,
> >     ) -> None:
> >         self.name = name
> >         self.dataset = dataset
> >         self.llm = create_llm_instance(llm_config)
> >         self.llm.cost_manager = CostManager()
> >         self.custom = operator.Custom(self.llm)
> >         self.custom_code_generate = operator.CustomCodeGenerate(self.llm)
> >         self.test = operator.Test(self.llm)
> >         self.sc_ensemble = operator.ScEnsemble(self.llm)
> >
> >     async def __call__(self, problem: str, entry_point: str):
> >         """
> >         Implementation of the workflow
> >         Custom operator to generate anything you want.
> >         But when you want to get standard code, you should use custom_code_generate operator.
> >         """
> >         # Use custom_code_generate to generate a solution
> >         generated_solutions = []
> >         for _ in range(5):  # Increased number of solutions for better ensemble choice
> >             solution = await self.custom_code_generate(problem=problem, entry_point=entry_point, instruction="")
> >             generated_solutions.append(solution['response'])
> >
> >         # Use ScEnsemble to select the most frequent solution
> >         ensemble_result = await self.sc_ensemble(solutions=generated_solutions, problem=problem)
> >         best_solution = ensemble_result['response']
> >
> >         # Validate the solution using the Test operator
> >         test_result = await self.test(problem=problem, solution=best_solution, entry_point=entry_point)
> >
> >         # If the test fails, get feedback and modify the solution accordingly
> >         if not test_result['result']:
> >             feedback_response = await self.custom(input=test_result['solution'], instruction="Provide feedback on errors and suggest corrections.")
> >             best_solution = await self.custom_code_generate(problem=problem, entry_point=entry_point, instruction=feedback_response['response'])
> >             # Re-test the new solution
> >             test_result = await self.test(problem=problem, solution=best_solution['response'], entry_point=entry_point)
> >
> >             # Additional feedback loop to improve upon the previous best solution
> >             if not test_result['result']:
> >                   # Generate suggestions using Custom
> >                   improvement_suggestions = await self.custom(input=best_solution['response'], instruction=prompt_custom.FEEDBACK_SUGGESTIONS)
> >                   best_solution = await self.custom_code_generate(problem=problem, entry_point=entry_point, instruction=improvement_suggestions['response'])
> >                   # Re-test the improved solution
> >                   test_result = await self.test(problem=problem, solution=best_solution['response'], entry_point=entry_point)
> >
> >         return test_result['solution'], self.llm.cost_manager.total_cost
> >  ```
> >  Workflow generated by W4S (MBPP)
> >  ```python
> > def workflow(agent, task: str, entry_point: str):
> >     instructions = "Requirements:\n1. Please explain your solution step by step.\n2. The answer MUST be a valid Python function.\n3. Use clear variable names and add comments for clarity."
> >     prompt = f"Your Task: \n{task}\nGenerate the complete function below with the function name equal to {entry_point}: "
> >
> >     messages = [{"role": "user", "content": prompt}]
> >     response = agent.call_json_format_llm(
> >         messages=messages,
> >         temperature=0.3,
> >         num_of_response=3,
> >         agent_role="Python Programmer",
> >         return_dict_keys=["reasoning", "answer"],
> >         instructions=instructions.strip(),
> >     )
> >
> >     return_dicts = response
> >     correct_solution = None
> >
> >     for return_dict in return_dicts:
> >         solution_code = return_dict.get("answer", "")
> >         results = agent.test_on_public_test(task, solution_code, entry_point, test_loop=3)
> >         if results['result']:
> >             correct_solution = results['solution']
> >             break
> >
> >     if correct_solution is None:
> >         # If no correct solution is found, take the first one
> >         correct_solution = return_dicts[0]['answer']
> >
> >     return_dict = {
> >         "answer": str(correct_solution),
> >         "reasoning": return_dicts[0].get("reasoning", ""),
> >     }
> >
> >     return return_dict
> >  ```

---

> > > ### Author Response · Authors · 2025-06-09
> > > **Response to Reviewer vbtG (3/n)**
> > >
> > > Workflow generated by ADAS
> > >
> > >  ```python
> > > def forward(self, taskInfo):
> > >     # Instruction for step-by-step reasoning
> > >     cot_instruction = "Please think step by step and then solve the task."
> > >     N = 5 # Number of CoT agents
> > >
> > >     # Initialize multiple CoT agents with a higher temperature for varied reasoning
> > >     cot_agents = [LLMAgentBase(['thinking', 'answer'], 'Chain-of-Thought Agent', temperature=0.8) for _ in range(N)]
> > >
> > >     # Instruction for final decision-making based on collected reasoning and answers
> > >     final_decision_instruction = "Given all the above solutions, reason over them carefully and provide a final answer."
> > >     final_decision_agent = LLMAgentBase(['thinking', 'answer'], 'Final Decision Agent', temperature=0.1)
> > >
> > >     possible_answers = []
> > >     for i in range(N):
> > >         thinking, answer = cot_agents[i]([taskInfo], cot_instruction)
> > >         possible_answers.extend([thinking, answer])
> > >
> > >     # Make the final decision based on all generated answers
> > >     thinking, answer = final_decision_agent([taskInfo] + possible_answers, final_decision_instruction)
> > >     return answer
> > >  ```
> > >  Workflow generated by AFlow
> > >
> > >  ```python
> > > XXX_PROMPT = """
> > > Solve the problem thoroughly using the thought process and the preliminary answer provided.
> > > How do you reach the conclusion based on the provided information?
> > > """
> > >
> > > REVIEW_PROMPT = """
> > > Carefully review the answer provided and consider any possible improvements.
> > > Does the answer fully address the problem with all necessary details?
> > >
> > > class Workflow:
> > >     def __init__(
> > >         self,
> > >         name: str,
> > >         llm_config,
> > >         dataset: DatasetType,
> > >     ) -> None:
> > >         self.name = name
> > >         self.dataset = dataset
> > >         self.llm = create_llm_instance(llm_config)
> > >         self.llm.cost_manager = CostManager()
> > >         self.custom = operator.Custom(self.llm)
> > >         self.answer_generate = operator.AnswerGenerate(self.llm)
> > >         self.sc_ensemble = operator.ScEnsemble(self.llm)
> > >
> > >     async def __call__(self, problem: str):
> > >         """
> > >         Implementation of the workflow
> > >         """
> > >         # Step 1: Generate thought process and preliminary answer
> > >         answer_data = await self.answer_generate(input=problem)
> > >         thought = answer_data['thought']
> > >         preliminary_answer = answer_data['answer']
> > >
> > >         # Step 2: Validate the preliminary answer for consistency
> > >         validated_answer_data = await self.sc_ensemble(solutions=[preliminary_answer])
> > >         validated_preliminary_answer = validated_answer_data['response']
> > >
> > >         # Step 3: Use the validated preliminary answer as part of the input for the custom method
> > >         consolidated_input = f"{problem}; Thought: {thought}; Prelim: {validated_preliminary_answer}"
> > >         custom_output = await self.custom(input=consolidated_input, instruction=prompt_custom.XXX_PROMPT)
> > >
> > >         # Step 4: Review and potentially revise the answer
> > >         review_input = f"{problem}; Thought: {thought}; Validated: {custom_output['response']}"
> > >         reviewed_output = await self.custom(input=review_input, instruction=prompt_custom.REVIEW_PROMPT)
> > >
> > >         # Step 5: Use ScEnsemble on possible answers to ensure consistency
> > >         answers = [validated_preliminary_answer, reviewed_output['response']]
> > >         consistent_answer = await self.sc_ensemble(solutions=answers)
> > >
> > >         return consistent_answer['response'], self.llm.cost_manager.total_cost
> > > """
> > >  ```

---

### Official Review · Reviewer_4Sc7 · 2025-05-12

**Rating:** 7
**Confidence:** 4
**Ethics Flag:** 1

**Summary:**

This work proposes a new framework (W4S) for optimizing LLM workflows that use weaker meta-agents to harness stronger models for execution. The system is built on two key concepts: 1) instead of a node-based workflow, W4S uses code generation with workflow interfaces (enabling more expressiveness) 2) instead of a training-free approach, or tuning a large model, W4S uses RLAO to tune a weaker model. W4S is evaluated on a range of reasoning and QA LLM benchmarks, comparing against hand-crafted workflows and training-free related work. W4S demonstrates superior performance to both across seen and unseen tasks. A cost analysis is given showing that W4S is more cost effective than training-free approaches ADAS and AFlow.

----
**update**

I have updated my score based on the response that provides further quantitative and qualitative evaluation that makes it easier to understand the types of workflows generated by W4S and the improvement with respect to related work.

**Questions To Authors:**

(115) How is w_0 chosen. How sensitive is the system to the initial workflow example?

(Table1) Is the SFT on 4o-mini using the same dataset as W4S w/ SFT, or a different dataset?
- If it is different, why not FT 4o-mini using the same dataset. Is it due to cost?
- If it is the same, could you elaborate on why W4S is better?

(Table2) Why does Table 1 use 4o-mini and Table 2 use 3.5 Turbo?

(Table3) AFlow has cheaper inference cost (0.3 vs 0.5) but much higher wall-clock time (10.9 vs 2.7). Could you explain why W4S is much faster? Is it due to parallelization in the workflow code?

(Workflow for GSMHard) The generated workflow uses the prompt: “The code must define a solution() function and return only the final numerical answer.” which matches the dataset description, but does not seem a requirement for the validation process (that just compares returned answer). How did the system learn to do this? Was it from the task description, or the pre-training or fine-tuning?

**Reasons To Accept:**

- Interesting and well-motivate problem domain.
- Novel implementation that improves over the selected baselines and related work.
- Clear presentation and writing.

**Reasons To Reject:**

The paper offers good results however it could be improved through more detailed analysis and explanation. Giving a stronger presentation for how the demonstrated improvements are achieved, and what the system is learning, would lift this paper and warrant a higher score.

*Unclear why the system is better:*

The W4S system describes two notable differences to related work (AFlow): a training approach using RLAO, and a code-based approach rather than graph-based. The presentation does not make it clear how code-based workflows improve the system. Are the resulting workflows produced by W4S inexpressible by AFlow, or are the resulting workflows simply more effectively instantiations of the available techniques/nodes?

What does the model learn through training? The generated workflow for MATH appears like a standard combination of CoT and Majority answer. How does it compare to the AFlow solution (why is it better by nearly 5% compared to AFlow and COT SC). Why does W4S beat AFlow by 24% on GSM Hard, but only  8% on MGSM)?

Broadly, it would be interesting to understand what contributes to the improvements. Is it optimization of hyperparameters, prompt wording, novel combinations of approaches, or stacking multiple approaches?

*Limited to well-known reasoning and QA tasks:*

The authors acknowledge this limitation and the results are compelling even within this domain. I have two main concerns:

As mentioned below, these are well studied datasets, and while solutions may not leak into training sets, approaches and workflow code, could. Is there a risk of this?

The authors present generalization results from the seen GSM Plus and MGSM to the unseen GSM8k, GSM Hard, and SVAMP. However, many of these datasets are similar to, or derivations of, GSM8k. Do these datasets offer significant differences and challenges with respect to the workflow code required to solve them, seeing as they are all math problems? A more detailed comparison would be helpful here. For instance, how does performance differ if the GSM Plus workflow is applied to GSM Hard, or vice versa? If there is a significant difference that would be a stronger indicator of differences between the datasets.

*Only one model is evaluated as trained weak meta-agent:*

Only one model is used (Qwen2.5-Coder-7B-Instruct) as a trained weaker-agent. To what extent does code or instruction tuning matter? Given that many of the datasets are publicly discussed in research papers, and LLM workflows for these datasets exist on GitHub (not the test sets themselves), is it possible that a model could be biased towards generating better workflows for these datasets via exposure during pre-training? Discussion of model choice or comparisons using other weak models would improve the results. As would discussion of whether exposure to existing workflow code or approaches could be an issue.

*‘Weak for strong’ narrative is unconvincing:*

In this work, the best results are achieved through a training approach not generally accessible on strong models (due to cost or API restrictions). However, if such a training framework was available, would we expect the resulting system to be better than one trained using a ‘weak’ or ‘weaker’ meta-agent?

Further, from a cost-argument, given that workflows employ strong models that can be executed many times once built, is it reasonable to expect that the overall cost of a system will be dominated by the workflow execution, with costs of workflow optimization amortized across usage?

I applaud the authors for demonstrating SOTA results using a weak meta-agent, and my concern here is not the lack of using strong meta-agents. My concern here is using terminology that was motivated by fundamental differences in a context that is driven by practical differences. For example, the labels (reward) used in W4S are the same throughout the system (the answer to the qa or reasoning tasks), rather than weak labels from humans for superhuman tasks. Is there a scientific reason that motivates the weak/strong distinction?
My concerns could addressed in multiple ways:
- Stronger cost analysis that demonstrates the gains of using a weaker meta-agent over a realistic usage scenario (not a benchmark evaluation).
- Reducing or clarifying the use of ‘Weak for strong’ which intentionally draws comparison to (Burns et al. 2024) but lacks the same fundamental motivation.

---

> ### Author Response · Authors · 2025-06-03
> **# Response to Reviewer 4Sc7 (4/n)**
>
> > Additional Questions from Reviewer:
>
> * **How is w_0 chosen? Sensitivity to initial workflow (w\_0)**:
>
>   *Answer:* $W_0$ is chosen by domain:
>
>   + MBPP, HumanEval:  CoT
>
>   + DROP, MMLU-Pro, GPQA: CoT, CoT SC
>
>   + Math benchmarks: CoT, CoT-SC, and a CoT-with-code variant that runs the derived expression in Python.
>
>   Compared with ADAS which utilizes nine initial workflows, W4S necessitates fewer instantiations (1-3). We here provide more results on sensitivity to $W_0$:
>
>   Table: Performance on DROP using different $W_0$
>   | $W_0$             | DROP |
>   | -----------         | ----- |
>   | CoT                  | 87.9  |
>   | CoT SC * 5      |  87.3  |
>   | CoT + CoT SC |  87.5 |
>
>   Table: Performance on MBPP using different $W_0$
>   | $W_0$       | MBPP |
>   | ----------- | ---- |
>   | CoT         | 86.8  |
>   | CoT SC * 5  | 84.8  |
>   | CoT w/ Test |  87.6   |
>   | CoT + CoT SC * 5 + CoT w/ Test | 87.9 |
>
>   Table: Performance on MGSM using different $W_0$
>   | $W_0$        | MGSM |
>   | ------------ | ---- |
>   | CoT           |   65.5   |
>   | CoT SC * 5          |   66.8   |
>   | CoT w/ code       |   62.5   |
>   | CoT + COT SC + CoT w/ code |  68.2 |
>
>   The reason why we seed math tasks with both a plain CoT template and a CoT-with-code template is to let the meta-agent sees syntactically correct examples of how to utilize LLM call helper function to get text or code. When we provided only plain CoT, the agent still attempts to add code generation, but its early scripts frequently crashed with syntax errors.
>
> * **(Table 1) Dataset used for SFT on 4o-mini vs. W4S w/SFT**:
>
>   *Answer:* Thanks for the question, but there might be a misunderstanding. In table 1, Finetuned GPT-4o-mini is trained on validation samples rather than serving as the meta-agent, so it's reasonable that the SFT tuned GPT-4o-mini falls short. However, it's true that we can finetune GPT-4o-mini as meta-agent, but the trajectories should not be the same as used for W4S with 7B Qwen since those trajectories are collected using 7B Qwen itself. We could use GPT-4o-mini as meta-agent to collect trajectories and filter out those with low rewards (since now we do SFT). Here we provide the corresponding experiments.
>
>   Table: Accuracy performance on different benchmarks. Executor is GPT-3.5-Turbo.
>   |                             | MGSM | GSM Plus | GSM Hard    | GSM 8k | SVAMP |
>   | --------------------------- | ---- | -------- | --- | -------- | -------- |
>   | ADAS                        | 57.4 |  53.4  |  34.5|   61.1 | 82.8 |
>   | W4S (7B Qwen)               |  68.2  |  **66.2**  | 61.8 | **86.5**  |    84.2  |
>   | W4S (Finetuned GPT-4o-mini) | **68.6** |   64.5   | **63.4** | 85.7  | **85.5**  |
>
>   Since finetuning API requires more API cost, training a weak model serve as more efficient and effective approach.
>
> * **Reasoning for different model use in Tables 1 and 2 (4o-mini vs. 3.5 Turbo)**:
>
>   *Answer:* We use different strong models as executors to show that our method performs well regardless of model use. We choose these two models because ADAS utilized 3.5 Turbo and AFlow utilizes GPT-4o-mini. We also provide results using GPT-4o and Claude as strong executors in the Appendix.
>
> * **(Table 3) Explanation for W4S's faster wall-clock time compared to AFlow**:
>
>   *Answer:* Yes, AFlow executes sequentially while ours benefit from parallelization.
>
> * **Workflow for GSMHard: Explanation of prompt choice**:
>
>   *Answer:* In this case, the meta-agent utilizes the provided helper function `agent.execute_code(code)`. This helper function has the following desciption shown in the prompt (can also check Appendix A.2):
>   *"`agent.execute_code(code)`: Execute the code and return the output. The code MUST contain a `solution` function. The output of `execute_code(code)` will be the return value of the `solution` function if the code is executed successfully or raise an exception."*
>
> ---
>
> We greatly appreciate the reviewer’s feedback and are confident these clarifications strengthen the paper significantly.

---

> > ### Comment · Reviewer_4Sc7 · 2025-06-04
> >
> > Thank you for the detailed response and additional evaluation. The extra quantitative and qualitive analysis helps to round-out the work and provide a better picture, and should be included in the paper (main or appendix as appropriate). I will increase my score in the next iteration.
> >
> > ---
> >
> > Some additional comments that the authors may want to consider. These are primarily focussed on presentation of the scoping of the work, not significant methodological flaws.
> >
> > > potential risk of solution leakage into workflow training
> >
> > The claim that the model has not seen workflow code and is not simply reciting templates is convincing. What is less clear is whether the system is identifying novel techniques or finding effectively ways to remix existing techniques. Translating a task before solving it is documented. The MGSM paper includes this variation:
> >
> > > Finally, we can translate the problem to English and solve it with English CoT (TRANSLATE EN).
> > (https://arxiv.org/pdf/2210.03057)
> >
> > From the presentation of the results in the submission, it is unclear whether W4S offers a way to discover novel workflows that a human would not think of on new tasks, or whether W4S is an effective "developer aid" - finding the most effective combination of techniques without needing the user to write any boilerplate. I'm inclined to believe it is the latter as all the benchmarks of of a similar flavour, vs a task such as writing a short story.
> >
> > This does not affect my score:
> > - Mathematical / reasoning tasks are a scoped but interesting and valuable domain to study.
> > - The improvements over baselines / models with larger/more recent training indicates that W4S does add new information.
> >
> > This could contribute to a discussion section, or make for follow-on work.
> >
> > > Would a trainable strong meta-agent be better? In principle, fine-tuning strong models as a meta-agent could yield marginally better workflows, since it might generate more fluent code or choose subtler prompt strategies.
> >
> > I agree with the argument here in the context of the benchmarks and datasets used. To my earlier point, the disparity between weak and strong meta-agents may be more pronounced on harder tasks (SWE-BENCH) or open-ended tasks (story writing). Interesting to think about, but I agree that this is unrealistic given the compute constraints.
> >
> > > If this wording remains confusing, we are open to alternatives (e.g., “Open-for-Closed”) that better reflect the practical constraints of unmodifiable, high-capacity LLMs.
> >
> > My updated score does not depend on any title change.

---

> > > ### Author Response · Authors · 2025-06-09
> > > **Response to Reviewer 4Sc7**
> > >
> > > Thank you for your thoughtful follow-up and insightful suggestions. We're delighted that the additional quantitative and qualitative analyses addressed your main concerns. We will incorporate these analyses into the revised manuscript.
> > >
> > > We appreciate your additional comments and agree with your assessment that rather than inventing fundamentally new techniques, W4S primarily excels at finding the most effective combinations of existing methods and optimizing prompts and hyperparameters automatically. We will clarify this explicitly in the paper and add a discussion section as you suggested.
> > >
> > > Your comments about the potential benefits of stronger meta-agents on harder or more open-ended tasks (e.g., SWE-BENCH, story-writing), are also valuable and insightful. For scenarios where a 7B model may be insufficiently capable, our approach naturally extends to stronger yet still relatively affordable (14B, 32B) models as meta-agents. These remain "weak" relative to large-scale executors, preserving practical efficiency while potentially unlocking more subtle workflow innovations. We believe these points offer compelling directions for future exploration.
> > >
> > > Thank you again for your constructive feedback, which has meaningfully improved our work.

---

> ### Author Response · Authors · 2025-06-03
> **# Response to Reviewer 4Sc7 (3/n)**
>
> > Q3: Only one model is evaluated as trained weak meta-agent:
>
> We thank the reviewer for pointing out this question. We here provide more experiments using different weak models as meta agents.
>
> Table: Testing performance with GPT-3.5-Turbo as Executor.
> |                                 | DROP | MGSM |
> | ------------------------------- | ---- | ---- |
> | COT                             | 64.2 | 28.0 |
> | ADAS (GPT-4o)                   | 79.4 | 53.4 |
> | AFlow (GPT-4o)                  | 78.5 | 54.8 |
> | W4S (Qwen2.5-7B-Coder-Instruct) w/o RLAO |     76.7 | 59.5 |
> | W4S (Qwen2.5-7B-Coder-Instruct) | 81.5 (+6.3%)   | 66.2 (+11.8%) |
> | W4S (Qwen2.5-7B-Instruct) w/o RLAO | 74.3 | 61.7 |
> | W4S (Qwen2.5-7B-Instruct)       |  83.7 (+12.7%)   |  66.3 (+7.5%) |
> | W4S (Qwen2.5-3B-Instruct) w/o RLAO | 67.8  |  57.5 |
> | W4S (Qwen2.5-3B-Instruct)       |  84.1 (+19.4%)   |  64.3 (+11.1%)  |
> | W4S (Llama3.1-8B-Instruct) w/o RLAO |  74.9 | 43.3  |
> | W4S (Llama3.1-8B-Instruct)  |  77.1 (+2.9%) |63.2 (+31.5%)  |
>
> ---
>
> > ‘Weak for strong’ narrative is unconvincing:
>
> Thanks the reviewer for the insightful question. We here provide detailed explanation to clarify your concerns.
>
> * **Contribution of novel approach and SOTA results:** Our primary contribution is the novel method using RL to train meta-agent to harness strong executors. Our SOTA results are meaningful contributions to the community. We will reduce emphasis on the "weak for strong" framing to avoid confusion.
>
> * **Would a trainable strong meta-agent be better?** In principle, fine-tuning strong models as a meta-agent could yield marginally better workflows, since it might generate more fluent code or choose subtler prompt strategies. In practice, however, two factors limit this advantage:
>   1. The meta-agent’s task is primarily workflow composition rather than deep problem solving. Once the agent can (i) write syntactically correct Python and (ii) reason about what techniques to use and how to improve, further raw language ability or reasoning abilities **yields diminishing returns**.
>   2. The most capable LLMs today—o3, Gemini 2.5-Pro, Claude—are not finetunable at all; even open-weight 70B checkpoints are unaffordable to train for most practitioners.
> * **Cost analysis in a realistic usage scenario.** It's true that the cost of generating workflows can be amortized, but this amortization is not as straightforward as it might seem. For real application, the execution of strong models typically uses API calls (even for a large open-sourced model), the cost will be the API credits. But to train a strong open-sourced model, one has to use unaffordable GPU compute along with training-time cost.
>
> * **Clarification or justification of 'Weak for Strong' terminology**: (Burns et al. 2024) used “weak-to-strong generalisation” for human labels supervising super-human tasks. Our setting is different yet analogous: a cheap, trainable LLM (weak) learns a policy-level skill that boosts a untrainable, frozen or even ununderstandable LLM (strong). The scientific question in both cases is the same: can limited supervision bootstrap a more powerful system than the supervisor itself? The question arises in Burns et al. (2024) because they focus on the analogy of humans supervising superhuman models; in our context, it emerges from the practical constraint of strong models being untrainable and increasingly opaque to human understanding as their capabilities grow. If this wording remains confusing, we are open to alternatives (e.g., “Open-for-Closed”) that better reflect the practical constraints of unmodifiable, high-capacity LLMs.

---

> ### Author Response · Authors · 2025-06-03
> **# Response to Reviewer 4Sc7 (2/n)**
>
> > Q2: Limited to well-known reasoning and QA tasks:
>
> * **Potential risk of solution leakage into workflow training**:
>
>   *Answer:* Here we clarify why our results are unlikely to come from the meta-agent merely “reciting” templates that appeared in its pre-training data:
>     1. Direct leakage of workflow code is impossible since the meta-agent never sees any explicit mapping of the form '<task, strong mode> --> workflow'.
>     2. Since the reward depends on relative improvement, simply reproducing a canned program offers no advantage once it has already plateaued; the policy is pushed to modify the previous script, not to replicate something it might have read in pre-training.
>     3. Previous work like ADAS and AFlow utilize GPT-4o or Claude as meta-agent, whose pre-training corpora are strictly larger than any open-weight 7B model, but they still underperform W4S when used inside ADAS or AFlow.
>
>
> * **Detailed comparison of dataset differences (GSM Plus, GSM8k, GSM Hard, SVAMP)**:
>   *Answer:* We here provide comparison of 5 datasets.
>
>
> | Dataset | Extra challenge  | Workflow differences          |
> | ------- | -------------------------------------------------------------------------------- | ------------------------------------------------------------------------------------------------ |
> | MGSM    | More difficult than GSM8k in terms of math reasoning + Multilingual descriptions | Translator -> Programmer (CoT text + code)*k → vote (if fails, fall back to role-play reasoning+SC) |
> | GSM Plus | Adversarial re-phrasings and distracting facts. The difficulty increases in the understanding of the questions as well as answering 'None' to unsolvable questions| Using both multiple roles of reasoning with text and Python programmer. |
> | GSM Hard | Same wording w/ GSM8k but numbers ×100–×1000 larger. Calculation using text fall short | emphasize coding answers, fall back to text reasoning only when all all generated code fails |
> | GSM8K   | Clean English; no adversarial noise. | Multiple role-play math reasoning (with detailed prompts to prevent calculation errors) * 5 combined with Programmer produced answers, all voting together; similar to GSM Plus |
> | SVAMP   | math word problems focused on arithmetics; relatively easy reasoning and calculations; programmer falls short because errors in code functions more often than errors in calculations with text | multiple role-play CoT with voting, no code generation |
>
> We here provide results using different workflows that are gerated for GSM Plus or GSM Hard.
>
> Table: Performance on GSM Plus and GSM Hard using different workflows. The Executor is GPT-3.5-Turbo.
> |                           | GSM Plus | GSM Hard |
> | ------------------------- | -------- | -------- |
> | Workflow 1 (for GSM Hard) | 64.2     | **66.6**     |
> | Workflow 2 (for GSM Hard) | 57.0     | 61.8     |
> | Workflow 1 (for GSM Plus) | **65.6**     | 57.2    |
> | Workflow 3 (for GSM Plus) | 64.4     | 64.8    |
>
>
> As shown in the table, the performance differs when the GSM Plus workflow is applied to GSM Hard, and vice versa. We will update this comparison as well as the detailed workflow code in our paper.

---

> ### Author Response · Authors · 2025-06-03
> **# Response to Reviewer 4Sc7 (1/n)**
>
> We thank the reviewer for the thoughtful and insightful comments that make our paper better. Below, we address each raised point individually.
>
> ---
>
> > Q1: Unclear why the system is better:
>
> **Clarification on code-based vs. graph-based workflows**:
>   *Answer:* Graph-based systems like AFlow use a fixed library of nodes (3 to 4 nodes for one task), each encapsulating hard-wired code and a fixed prompt. The meta-agent cannot modify a node’s internal logic or introduce entirely new modules. As [1] already observed, its action space is narrow and prone to premature convergence on suboptimal templates.
>
>   By contrast, W4S treats workflow design as a programming task: the meta-agent writes arbitrary Python code that can call strong models, combine steps, and implement error handling on the fly:
> - **Unrestricted module creation**. The agent can implement unrestricted techniques by calling strong models. On MMLU-Pro it spontaneously spawned role-specialised sub-agents (“scientist”, “reason expert”, etc) by specialised system prompt and fused their answers. On multilingual task, the meta-agent happens to create a translator module that translates original prompt to English version. Besides, for mathematical tasks like GSM Hard, the meta-agent introduces error handling of code execution.
> - **Editable implementations of the same primitive**. Even for the same techniques like “CoT SC”, W4S can re-implement it. For example, on MATH, the agent discovered that extracting the final numbers from each chain and doing simple majority-voting outperforms using ensemble which feeds the entire reasoning trace into another LLM call. The latter is what AFlow’s built-in Ensemble node does. But on benchmarks like DROP, the meta-agent learn to use ensemble instead of majority voting of final answers.
> - **Granular prompt and hyper-parameter control**. Temperatures, sample counts, agent roles (system prompts) and etc are micro-adjustments in Python that contributes to improvement.
>
> Thanks for pointing out this question, we will include the comparison and analysis to appendix of the paper.
>
> [1] ScoreFlow: Mastering LLM Agent Workflows via Score-based Preference Optimization
>
> **What does the model learn during training (MATH example)**: As mentioned in the last question, for MATH dataset, the meta-agent in W4S learns that COT with final answer majority voting performs better than COT with ensemble (which is conventional COT SC does), it also learns that directly solving the problem performs better than python code generation since the answers in MATH are often LaTex expressions. Below is the best workflow generated by AFlow within 20 iterations for MATH, which is a conventional COT prompting followed by python code generation.
> ```python=
> XXX_PROMPT = """
> # Please provide a detailed solution to the following problem, ensuring each mathematical step is explained.
> #
> # Solve it.
> #
> # """
> class Workflow:
>     def __init__(
>         self,
>         name: str,
>         llm_config,
>         dataset: DatasetType,
>     ) -> None:
>         self.name = name
>         self.dataset = dataset
>         self.llm = create_llm_instance(llm_config)
>         self.llm.cost_manager = CostManager()
>         self.custom = operator.Custom(self.llm)
>         self.programmer = operator.Programmer(self.llm)
>
>     async def __call__(self, problem: str):
>         """
>         Implementation of the workflow
>         """
>         # Step 1: Obtain initial solution using Custom operator
>         custom_response = await self.custom(input=problem, instruction=prompt_custom.XXX_PROMPT)
>         initial_solution = custom_response['response']
>
>         # Step 2: Cross-verify with Programmer operator
>         programmer_response = await self.programmer(problem=problem, analysis=initial_solution)
>         final_answer = programmer_response['output']
>
>         return final_answer, self.llm.cost_manager.total_cost
> ```
>
> **GSM Hard vs. MGSM performance differences**: (1) MGSM is a multilingual math task that also requires multilingual abilities. As shown in Figure 4, the case-study workflow for MGSM and followed by code generation (2) GSM-Hard replaces the numbers in GSM8k with much larger ones, turning questions into pure long-arithmetic challenges where resampling and prompting methods all fail but do benefit from code execution. The workflow generated for GSM-Hard utilizes code generation with error handling mechanisms, which improves the performance largely.
>
> **Contributing factors to performance improvements**: The performance improvements stem from detailed prompts, hyperparameters (number of samples, temperatures, etc), suitable combinations of different techniques, different agent role playing (implemented with system prompt), different implementations of the same techniques.

---

### Official Review · Reviewer_iWtW · 2025-05-30

**Rating:** 6
**Confidence:** 3
**Ethics Flag:** 1

**Summary:**

This paper proposes the novel idea of using a weak model to generate workflows for a strong model to execute and using offline RL to train this weak model for specific tasks. The authors present this framework as an alternative to training the strong model. They evaluate their framework on eleven benchmarks, covering math-reasoning, question-answering, and code generation. Their framework achieves superior performance across the benchmarks compared with the strongest workflow optimization baselines while being more efficient in time and cost in both workflow generation and execution.

**Questions To Authors:**

1. Have you tried putting different caps on the inference-time compute allowed for the executor? Maybe it’s not happening in your experiments but is it possible that some meta-agent would learn to just let the executor scale up inference-time compute?

2. Do the selected tasks adequately reflect the real-life tasks that require the strong model to solve? Can some of them be solved by the weak coding model directly?

**Reasons To Accept:**

1. The task formulation of using a weak model to generate workflows so that training can be done just on the weak model is novel.

2. The superior efficiency in both workflow optimization and execution compared with other workflow optimization baselines show the promise of the overall approach in real-life applications.

**Reasons To Reject:**

1. Lacks baselines that directly train the strong model. The only such baseline, finetuned GPT4o-mini, just uses supervised finetuning on the validation set, so it’s not surprising that the results are unsatisfactory. The authors should also try other training methods for the strong model, such as more data-efficient approaches and common algorithms of online and offline RL. Other open-sourced strong models should easily support different training algorithms.

2. Using a meta-agent alone without RLAO training already improves performance on several benchmarks, and the improvements from RLAO seem marginal on DROP, MMLU pro, MBPP, and HumanEval.

3. In Table 2, the “unseen” math tasks seem similar to the seen tasks, so it’s unclear whether the experiments can serve as evidence of generalizability.

4. Lacks comparison between the RLAO algorithm and other algorithms used for improving LLM coding and tool use. It would be helpful to see discussions on which parts of RLAO are novel.

5. The experiments only use one model as the weak meta-agent, so it’s unclear if the method generalizes to other weak models. Why did you choose Qwen2.5-Coder-7B-Instruct? Are there anything specific about this model that makes the framework work well?

---

> ### Author Response · Authors · 2025-06-03
> **# Response to Reviewer iWtW (3/n)**
>
> > Q6: inference-time compute
>
> We enforce a hard cap during execution. If a run exceeds the budget, execution is aborted and the trial is skipped. However, we did not explicitly penalise cost in the reward, so it's possible that the meta-agent learns to scale up inference-time compute. Therefore, we conduct cost analysis which demonstrates that our method does not require more inference-computing than baselines. Incorporating cost into framework could be an interesting future step to enhance the application.
>
> > Q7: Can datasets solved by the weak coding model directly?
>
> We follow previous automated workflow generation work like ADAS and AFlow to choose the datasets used in our paper (e.g., DROP, MATH, MBPP, etc). As mentioned in limitation section, implementing more complex tasks where weak models cannot solve at all will be a interesting future step. Besides, here we provide experiments on a quite difficult dataset AIME that the weak coding model cannot solve at all. We use AIME23 as validation set and test on AIME24.
>
> | Dataset (Executor)        | AIME (GPT-4o) |  AIME (GPT-4.1-mini)   | AIME (o3-mini-medium) |
> | ------------------------- | ------------------- | --- | --------------------- |
> | COT                       |   10.0   | 40.0 |   70.0  |
> | W4S (Qwen2.5-7B-Instruct) |   23.3   | 70.0  |  90.0  |

---

> > ### Comment · Reviewer_iWtW · 2025-06-05
> > **Thank you for your responses!**
> >
> > Thank you for your responses! They answered several of my questions.
> >
> > I still have a couple of follow-up comments.
> >
> > 1. Regarding the novelty of RLAO, I agree that it's a novel application of existing RL concepts (RWR, best-of-n, etc.) However, ideas such as computing rewards based on execution feedback or collecting trajectories offline have been widely used. So it's unclear whether bullet points 2-3 can be claimed as novel.
> >
> > 2. I’m still concerned about the lack of baselines that directly train a single model to solve the tasks. To clarify, I’m not asking you to train a model on the full training set. You can use the same dataset for RLAO and compute feedback similar to your current approach. You mentioned that training a strong model might be impractical (either because the model is proprietary or because of compute constraints). What about using different algorithms to train a model that has a size similar to the model you use as the meta-agent (e.g., 7b-9b)?
> >
> > 3. The additional AIME task is still too similar to a few of the existing tasks in the paper, so I’m still not sure about the conclusion on generalizability and what real-life scenarios this method may be suitable for. Overall, the paper will be much stronger with more diverse tasks and an explanation of why the tasks represent real-life scenarios where the proposed method is preferred over the baselines.

---

> > > ### Author Response · Authors · 2025-06-10
> > > **Response to Reviewer iWtW**
> > >
> > > Thank you again for the valuable feedback. Below we provide additional clarifications and experiments to address your remaining concerns:
> > >
> > > ---
> > > 1. Novelty of RLAO: We agree that the idea of computing rewards from feedback is not fundamentally new. Our novelty lies in combining this idea in the specific context of weak agents optimizing workflows since prior RL applications typically optimize a model's own capabilities directly.
> > > 2. We understand your concerns on lack of results of models directly trained on the task. We use GRPO to train Qwen2.5-7B-Instruct on GSM Hard (validation split) and test on testing samples. We acknowledge that on simple tasks like GSM8k, directly training an open-weight model can outperform W4S using strong executors. However, when task complexity and the capability gap between strong executors and weak models increase, leveraging a strong executor remains more beneficial than directly training a similarly-sized open-weight model, especially given limited compute or data. While fully training the strong executor is expected to surpass W4S, our approach provides a valuable alternative when such training is impractical due to cost or model availability constraints.
> > >
> > > |                                      | GSM Hard |
> > > | ------------------------------------ | -------- |
> > > | Qwen-7B-Instruct (untrained,COT)     |  32.2        |
> > > | Qwen-7B-Instruct (trained with GRPO) |  52.8 |
> > > | GPT-4o-mini (untrained,COT)                                      |     39.5     |
> > > | GPT-4o-mini (W4S)       | 76.6   |
> > >
> > > 3. To demonstrate broader generalizability and applicability to realistic agentic tasks beyond mathematics, we evaluated W4S on GAIA-text, a well-established agentic benchmark encompassing diverse domains like web browsing, file reading, PDF/image/video parsing, calculation, and more.
> > >
> > > Table: Performance on GAIA using GPT-4.1 as executor.
> > > |      | Avg  | Level 1    | Level 2 | Level 3 |
> > > | ---- | ---- | --- | --- | -------- |
> > > | CoT  | 18.1 | 28.57 |   13.6  | 10.5  |
> > > | CoT SC | 18.1 | 28.57 | 13.6 | 10.5 |
> > > | Self-Refine | 21.3 | 28.6 | 19.7 |  10.5|
> > > | LLM Debate|23.6| 35.7|18.2 |15.8 |
> > > |ADAS | 22.0 | 18.2 | 31.0 | 15.8 |
> > > | W4S  |  33.9   |   45.2  |   31.8  |  15.8   |
> > >
> > > ---
> > > We thank the reviewer for raising these questions. We will include the additions into our paper.

---

> > ### Comment · Reviewer_iWtW · 2025-06-11
> > **Thank you for the response!**
> >
> > Thank you for the follow-up! I'll keep my current score, which is positive. I'm not increasing the score primarily because my Concern 2 remains and because of the number of important experiments that were not present in the submitted version of the paper. I really appreciate the additional results from new experiments provided during the discussion period. I do think that you will have a good paper once all the complete results and any promised statistical tests from those new experiments are included.

---

> ### Author Response · Authors · 2025-06-03
> **# Response to Reviewer iWtW (2/n)**
>
> > Q3: "unseen" math tasks seem similar to the seen tasks
>
> - **Generalization beyond mathematics**: Table 1 already reports transfer from the same meta-agent to different domains, where W4S still outperforms automated baselines.
> - **"unseen" math tasks are genuinely different:** Although in table 2, all the tasks are basically math ones, they have subtle differences. e.g.(1) MGSM is a multilingual math task that also requires multilingual abilities. As shown in Figure 4, the case-study workflow for MGSM **begins with a Translator module, which never appears for the other datasets** (2) GSM-Hard replaces the numbers in GSM8k with much larger ones, turning questions into pure long-arithmetic challenges where resampling and prompting methods all fail but do benefit from code execution. GSM-Plus perturbs and re-phrases GSM8K items, introducing distracting facts and adversarial choices; success relies on logical robustness rather than large-number arithmetic, where solely relying on python calculator is not enough. The resulting workflows demonstrate differences and confirm adaptation.
>
> We will update a more detailed qualitative analysis and comparison with different generated workflows in camera-ready version.
>
> > Q4: Lacks comparison between the RLAO algorithm and other algorithms used for improving LLM coding and tool use.
>
> Thanks for your suggestion! We here provide detailed comparison and analysis on why RLAO is novel. The discussion here will be updated in later version of our paper.
>
> **Comparison with other algorithms**: Unlike RLHF, which depends on human‑labeled preferences or a learned reward model, RLAO computes rewards directly from a strong model’s validation scores—no separate reward network is required. Unlike “RL with verifiable rewards,” which optimizes absolute task accuracy, RLAO reinforces any workflow change that yields relative improvement, regardless of final performance. Methods such as Toolformer, ReAct, and LATS teach tool‑calling through self‑supervision or prompt engineering but do not learn a multi‑turn policy to refine workflows.
>
> **Key novelty of RLAO:**
>
> 1. **novel application of RL**: Rather than optimizing a model’s raw task‑solving ability, RLAO optimizes the weak agent’s skill at understanding the task and generating and improving workflows that call the strong model.
> 2. **Rewards are computed by comparing strong‑model validation performance** before and after each proposed workflow, eliminating the need for human preferences or an explicit reward network.
> 4. RLAO is multi-turn algorithm. And the training is entirely offline: since executing the strong model is expensive, RLAO avoids inefficient online updates (e.g., PPO) by using a batch of recorded workflows and their validation outcomes.
> 5. **Best-of-m selection**: accelerates convergence by focusing learning on the most promising trajectories in each iteration; enriches the trajectories by including both good and bad attempts
>
> In summary, RLAO is novel in what it optimises (ability to generate and improve workflow), how it learns (offline reward-weighted RL), and where it is applied (weak agents steering much stronger executors)—a setting not addressed by existing coding or tool-use RL methods.
>
> > Q5: Unclear if the method generalizes to other weak models
>
> Our framework is model-agnostic; it treats the weak meta-agent as a black-box policy that produces workflow code, then refines that policy with RLAO. We initially reported Qwen2.5-Coder-7B-Instruct because it's one of open-weight 7B models that has been post-trained on code, which makes our training more efficient and effective. Weaker models may spend more time on correcting format or syntax errors. To further prove the efficacy of our method, we here provide experiments with another three weak models.
>
> Table: Accuracy Performance of W4S using different models as meta-agents. The executor is GPT-3.5-Turbo
> |                                 | DROP | MGSM |
> | ------------------------------- | ---- | ---- |
> | COT                             | 64.2 | 28.0 |
> | ADAS (GPT-4o)                   | 79.4 | 53.4 |
> | AFlow (GPT-4o)                  | 78.5 | 54.8 |
> | W4S (Qwen2.5-7B-Coder-Instruct) | 81.5 | 66.2 |
> | W4S (Qwen2.5-7B-Instruct)       | 83.7 |   66.3 |
> | W4S (Qwen2.5-3B-Instruct)       | 84.1 |   64.3 |
> | W4S (Llama3.1-8B-Instruct)      | 77.1 | 63.2 |
>
> All four weak models outperform both strong-model optimisers (ADAS/AFlow) on at least one benchmark, demonstrating model-agnostic generalisation. Llama performs worse than other models. However, we currently stop after a single offline-RL pass; additional collect-and-retrain iterations will close the gap further.

---

> ### Author Response · Authors · 2025-06-03
> **# Response to Reviewer iWtW (1/n)**
>
> We are really grateful to the reviewer for the thoughtful and insightful comments to make our paper better. Below, we provide new experiment results and address each raised point individually.
>
> ---
>
> > Q1: Comparison with training strong models w/ RL
>
> The reasons why we did not compare with directly training strong models on a training dataset with more advanced training algorithms is that
> (1) Since we only utilize a small validation set, it's unfair to compare our method with strong models trained on full training dataset.
> (2) We acknowledge that with proper training (e.g., RL), trained strong models are expected to outperform W4S. However, our objective is not to outperform a fully-trained strong model, but to approach that upper bound when full training strong models is  impractical or even impossible. This is motivated by common real-world cases when commercial models are closed-sourced or comparable open-source models demand prohibitive compute.
>
> > Q2: meta-agent w/o RLAO already improves performance
>
> **1. Why the “raw” W4S meta-agent is already strong on some datasets**
>
> Even without RLAO, W4S has intrinsic advantages over existing baselines.
>
> - **Unconstrained code-represented workflows**. Unlike AFlow (graph-level, only 3 or 4 nodes for each task) that suffers from early convergence to sub-optimal, hard-wired templates, W4S lets the meta-agent write arbitrary Python code that calls the strong model to complete the task. This vastly enlarges the search space and encourage creativity.
> - **Short, information-dense context**. W4S maintains a short, information‑dense history window rather than accumulating a long transcript of past iterations (as ADAS does). By focusing the agent’s attention on the most recent, salient workflows—augmented with case studies of past failure modes—W4S avoids overwhelming its context with irrelevant history (we show in Appendix E.1 that **ADAS performs no better than random sampling**).
> - **Self-correction mechanism**. This ensures that simple coding errors do not derail the search: if a generated workflow fails on a single validation example, the meta‑agent is prompted to fix it up to three times. In contrast, AFlow lacks any error‑handling, so mistakes tend to propagate through subsequent iterations.
> - Besides, we utilize code-tuned weak model Qwen2.5-Coder, which exhibits relatively good coding ability.
>
> These factors let W4S beat baselines “out of the box.” on some benchmarks.
>
> **2. What RLAO still adds and experiments with different weak models**
>
> - **RLAO amplifies weaker agents**: To further prove the effectiveness of RLAO, we here provide more experiment results using different weak models as meta-agent. It's evident RLAO improves performance of all weak models and is more effective when the weak model is weaker (Llama, 3B Qwen). This confirms that RLAO behaves like a proper policy‑improvement algorithm: the weaker the initial policy, the more room there is to improve.
>
> Table: Accuracy Performance of W4S using different models as meta-agents. The executor is GPT-3.5-Turbo
> |                                 | DROP | MGSM |
> | ------------------------------- | ---- | ---- |
> | ADAS (GPT-4o) |79.4 | 53.4 |
> | W4S (Qwen2.5-7B-Coder-Instruct) w/o RLAO |     76.7 | 59.5 |
> | W4S (Qwen2.5-7B-Coder-Instruct) | 81.5 (+6.3%)   | 66.2 (+11.8%) |
> | W4S (Qwen2.5-7B-Instruct) w/o RLAO | 74.3 | 61.7 |
> | W4S (Qwen2.5-7B-Instruct)       |  83.7 (+12.7%)   |  66.3 (+7.5%) |
> | W4S (Qwen2.5-3B-Instruct) w/o RLAO | 67.8  |  57.5 |
> | W4S (Qwen2.5-3B-Instruct)       |  84.1 (+19.4%)   |  64.3 (+11.1%)  |
> | W4S (Llama3.1-8B-Instruct) w/o RLAO |  74.9 | 43.3  |
> | W4S (Llama3.1-8B-Instruct)  |  77.1 (+2.9%) |63.2 (+31.5%)  |
>
> - **Faster convergence and higher stability**: As shown in Figure 3, W4S w/ RLAO is more stable with faster convergence, leading to sequential improvements compared with baselines and untrained versions.
>
> **3. Take-aways**
> The baseline-beating performance without RL is a design achievement, not evidence that RLAO is unnecessary. RLAO still provides (i) consistent extra accuracy on high-ceiling tasks, (ii) large relative gains for truly weak models, and (iii) faster, more stable optimisation – all with <1 GPU-hour of training. We will add the expanded ablation table and statistical tests to the camera-ready version.

---

### Author Response · Authors · 2025-06-03
**General Response**

Dear Reviewers,

We sincerely thank all reviewers for their time, insightful reviews, and constructive suggestions. We are heartened that reviewers found our work to be well-motivated (R-4Sc7, R-vbtG), novel (R-iWtW, R-4Sc7, R-vbtG), supported by solid experiments (R-iWtW, R-vbtG), well-written (R-4Sc7), and efficient (R-iWtW, R-vbtG).

We appreciate the valuable feedback and have prepared detailed responses to each reviewer. Here, we first address some common concerns:

### 1.  Clarification on "Weak-for-Strong" Terminology and Motivation (R-4Sc7, R-vbtG):

Our core motivation is to develop methods for efficiently harnessing powerful but often black-box or expensive-to-tune "strong" executor models using smaller, adaptable "weak" meta-agents.

**Relative Strength:** As clarified in our response to R-vbtG, for the MATH benchmark, our evaluation focused on the most difficult problems (level 5), where the Qwen2.5-7B performance provided by the reviewer is on the full dataset. Besides, When using truly frontier models like GPT-4o as executors (see new AIME results in response to R-vbtG, and Appendix E.2), the distinction in capability is clearer.

**Scientific Question:** Our work explores whether a less capable (but trainable) model can learn to effectively orchestrate a more capable (but fixed or costly to train) model. This question is pertinent given the increasing prevalence of powerful API-based models. While the motivation differs from Burns et al. (2024) (human labels for superhuman tasks vs. weak LLM for strong LLM orchestration), we see an analogy in the shared scientific question: can limited supervision/capability bootstrap a system more powerful than the supervisor/orchestrator itself?

**Revisions:** We will refine the "Weak-for-Strong" narrative in the paper to more precisely reflect our setting and contributions. We will also reduce the emphasize of it as suggested by R-4Sc7.

### 2.  Generalization to Other Weak Meta-Agent Models (R-iWtW, R-4Sc7):

W4S is designed to be model-agnostic regarding the choice of the weak meta-agent.

**New Experiments:** In response to R-iWtW and R-4Sc7, we presented new experimental results using three additional weak models as meta-agents: Qwen2.5-7B-Instruct, Qwen2.5-3B-Instruct, and Llama3.1-8B-Instruct. These results (e.g., Table in response to R-iWtW Q2 & Q5, and R-4Sc7 "Only one model") demonstrate that:

- All tested weak models, when trained with W4S, outperform strong baselines like ADAS and AFlow on several benchmarks.
- Our RLAO training consistently improves the performance of these diverse weak meta-agents, with more significant gains observed for initially weaker models (e.g., Llama3.1-8B, Qwen2.5-3B). This confirms the robustness and general applicability of W4S.

### 3. Qualitative and Quantitative Analysis of W4S's Advantages (R-4Sc7, R-vbtG):

W4S's advantages, learning capabilities, and workflow distinctions include:
-    **Superior Expressiveness (R-4Sc7):** Unlike graph-based systems (e.g., AFlow) with fixed nodes, W4S's code-based workflows allow the meta-agent to generate arbitrary Python, create novel modules (e.g., "Translator" for MGSM), dynamically adjust parameters, and implement complex logic.
-   **Learned Task-Specific Strategies (R-4Sc7, R-vbtG):** Through RLAO, the meta-agent learns nuanced strategies:
    -   On MATH: Discovered optimal answer extraction (majority voting over ensembling) and preferred direct solving for LaTeX.
    -   On GSM Hard/MGSM: Adapted by prioritizing code execution with error handling (GSM Hard) or introducing translation steps (MGSM).
    -   Effectively utilized and adapted helper functions to task demands.
-   **Adaptive and Complex Workflows (R-4Sc7, R-vbtG):** W4S generates workflows that are more complex and adaptive than static prompts or AFlow's templates, uniquely combining techniques (CoT, code execution, self-correction, voting) tailored to the task and executor.
-   **Robustness through Failure Mitigation (R-vbtG):** W4S addresses failure modes like validation overfitting (via public/private splits), exploration collapse (best-of-m, model diversity), and syntax errors (self-correction).
-   **Generalization and Minimized Leakage (R-4Sc7, R-iWtW):** W4S demonstrates generalization to distinct "unseen" tasks. Its dynamic, reward-driven workflow generation encourages novel solutions over reciting pre-trained data, mitigating leakage concerns.

### 4. More difficult benchmarks and stronger executors (R-iWtW, R-vbTG)

In response to R-iWtW and R-vbTG, we provide new experiments on competition-level dataset **AIME** using **GPT-4o, GPT-4.1-mini and o3-mini-medium** as stronger executors. Results demonstrate that W4S continues to deliver tangible gains on more demanding tasks and **remains effective when paired with today’s top models**.

---

### Comment · Area_Chair_TbLh · 2025-06-09
**Discussion period ends in 2 days**

Dear Reviewers, please engage in the discussion with the authors. The discussion period ends in 2 days. Kindly read their responses and leave a comment mentioning whether you find their answers convincing and adjust score accordingly or if further concerns remain. Thank you!

---

### Decision · Program_Chairs · 2025-07-08

**Decision:**

Accept

**Comment:**

This paper presents W4S, a framework where a small meta-agent generates workflows to leverage stronger untrainable API-based LLMs. By combining code-based workflow generation with offline reinforcement learning and best-of-m action selection, W4S enables efficient orchestration of powerful models. The method is evaluated across 11 benchmarks—including math, QA, coding, and the GAIA agentic benchmark—consistently outperforming state-of-the-art baselines like ADAS and AFlow. Key strengths include a strong empirical results, efficiency and broad generalization. It excels at automating and optimizing workflow composition rather than inventing entirely new prompting strategies. Overall, this is a well-executed contribution and I recommend acceptance.